# A Collaborative Trans-Regional R&D Strategy for the South Korea Green New Deal to Achieve Future Mobility

**Doyeon Lee and Keunhwan Kim ***

Division of Data Analysis, Korea Institute of Science and Technology Information (KISTI), 66, Hoegi-ro, Dongdaemun-gu, Seoul 02456, Korea; dylee@kisti.re.kr
\* Correspondence: khkim75@kisti.re.kr

**Abstract:** In response to the COVID-19 pandemic, South Korea is moving to establish a national industry strategy to reduce regional inequalities within the country through the Green New Deal. Thus, it is important to closely integrate the aim of reducing greenhouse gas emissions from the Green New Deal with that of reducing deepening regional inequality from the Regionally Balanced New Deal. To accomplish these dual aims, this study provides a collaborative trans-regional R&D strategy and a precise framework with three key dimensions: regional, technological, and organizational. We demonstrate that future mobility is the most important project of the Green New Deal, comprising 1963 nationally funded projects worth USD 1285.4 million. We also illustrate the level of government investment in nationally funded research projects related to future mobility for 17 different regions and seven different technology clusters related to future mobility, and determine which research organizations played an important role in each cluster for all 17 regions between 2015 and 2020. Our results indicate that the capital region and Daejeon have high innovation capability in many future mobility-related research fields, whereas some regions have capabilities in specific research fields such as hydrogen infrastructure, indicating their relative competitiveness.

**Keywords:** Korean Green New Deal; collaboration; trans-regional R&D strategy; future mobility; nationally funded project data; framework



## 1. Introduction

In recent years, diplomatic conflicts and economic tensions, motivated by trade wars between the US and China over unfair trade practices and between South Korea (hereafter Korea) and Japan over their historical legacies, have greatly endangered trade and value chain linkages between these countries and had a substantial influence on the Korean economy, which is characterized by integrated trade and production networks [1,2]. Moreover, the ongoing global COVID-19 pandemic caused a major global recession, referred to as the Great Shutdown, which aggravated global value chain (GVC) disruptions [3,4]. Factories in Korea have closed not only because of lockdown measures but also because of a lack of domestic goods and halted supplies of parts and components from abroad. Therefore, Korea's value chains are likely to become less global and more regional in scope, owing to mounting uncertainty over cross-border transactions [5]. Thus, GVC disruptions caused by recent trade restrictions and the global COVID-19 pandemic are posing new challenges to the traditional macroeconomic and industrial policies of the Korean government [1].

Various developed nations, including the US (February 2019) and the EU (December 2019), have recently relaunched Green New Deals similar to the climate-oriented economic stimulus policies implemented after the 2008–2009 Great Recession; such green stimulus packages not only raise investments with short-term benefits for economic output and jobs, but also lay the groundwork for long-term innovation and economic development aligned with environmental constraints [6–9]. The Korean government also considered reintroducing the Korean Green Growth Initiative of 2009 in response to the COVID-19 crisis as a national industrial strategy to promote green innovation and transform the

industrial structure of key global industries such as motor vehicles, batteries, and electricity distribution systems [8,10]; the aim was to make Korea a competitive leader in the future global economic structure. Eventually, the Korean government announced the Green New Deal as one of the three pillars of the Korean New Deal on 14 July 2020, and proposed a total investment of KRW 73.4 trillion (KRW 42.7 trillion from the treasury) over the next five years [11].

Rising inequality within countries—that is, between prosperous metropolitan regions and less prosperous regions, and between core areas and peripheral areas—has become not only an economic problem but also a source of social and political instability in most developed and developing nations [12–15]. Korea is among the most geographically heterogeneous countries, where negative externalities have arisen from a heavily concentrated population in a single region and a drain of talents and resources from elsewhere, generating strong regional inequality [16,17]. The capital region (hereafter CR), defined as the jurisdiction of Seoul, Incheon, and Gyeonggi Province, contributes almost half of the national gross domestic product (GDP) (51.8%) and accounts for a similar share of the national population (50%), with Seoul alone representing 22.2% of the national GDP and 18.8% of the national population in 2019 [18]. The issue of regional inequality has prompted greater national demands for public investment since the economic depression resulting from the COVID-19 pandemic [19]. Consequently, the Regionally Balanced New Deal was announced as the third pillar of the Korean New Deal on 13 October 2020, with a projected investment of KRW 50.8 trillion (KRW 26.9 trillion from the treasury) [20]. Investment funds to support these New Deal projects are expected to be derived from both public and private sources.

Several studies have devoted to the normative statement that the Korean government endeavors to accomplish both the goal of reducing greenhouse gas emissions via the Green New Deal and that of reducing deepening regional inequality via the Regionally Balanced New Deal through effective collaborations with other stakeholders such as private companies, academia, research institutes, and agencies, owing to inter-linked concerns over climate change, air pollution, energy security, and the global competitiveness in the key industries [8,10,21]. However, studies on developing a systematic framework for collaborative trans-regional R&D strategy planning are lacking.

In an attempt to bridge this gap, we propose a systematic framework that would support the organization committee that consists of stakeholders who implement the national green industrial strategy by providing valuable and detailed information on relevant evidence-based situations and inclusive, relevant local implementation actors who may become the collaborative partners and/or members of the committee in a particular technology sector. Moreover, we applied the proposed framework on future mobility, one of the five key projects of the Green New Deal, to induce the implications for a collaborative trans-regional R&D strategy plan. Meanwhile, we investigated the global changes in public transport stemming from the COVID-19 pandemic to discuss the directions to the development of future mobility.

### 1.1. Theoretical Background and Literature Review

1.1.1. Inherent Purpose of Korean Green New Deal

Companies determine the values of products and/or processes based on research and development (R&D) activities; therefore, technological innovation is treated as one of the main determinants of total factor productivity, profit, and economic growth [22–24]. Thus, regional inequality results from variations in the technological and scientific resources required to achieve critical mass and develop sufficient absorptive capacity to participate in the dynamics of global science-led and R&D-based innovation [25,26]. Regional income/wealth inequality has increased substantially and steadily without signs of improvement in many countries [14]. In accordance with the global trend of regional inequality, there is growing regional inequality in the Seoul metropolitan area and Chungcheong provinces in terms of the amount of investment and human resources for technology R&D

in Korea. Moreover, public R&D, which should rectify the imbalance in private R&D investment, is also concentrated in Seoul and Daejeon (more than 80% [18]). However, the concentration of core technical personnel in the CR of Korea is even more serious than the imbalance of R&D investment, with the technological workforce continually concentrating in this region over the last 10 years [1,16].

Thus, several of the world's leading innovation economies, such as the US and EU, are moving to establish innovation strategies or policies that reduce inequalities within countries or between countries through Green New Deals [6,8,27]. The focus of such innovation strategies or policies is to promote the exchange of knowledge and other assets within and beyond regions [28–31]. Therefore, the involvement of various stakeholders, actor networks, and policy agents in different regions is ensured by establishing communication channels for horizontal and vertical coordination [25,32,33]. Although the Korean Green New Deal was not specifically designed to alleviate the issue of regional inequality in Korea, the central government made a commitment with local governments to implement key innovative projects [11]. Thus, it is important to closely combine the aim of reducing greenhouse gas emissions (from the Korean Green New Deal) with that of reducing deepening regional inequality (from the Regionally Balanced New Deal) by establishing a national collaborative R&D strategy for a trans-regional innovation approach that will accelerate the creation, dissemination, absorption, and application of new scientific and technological knowledge and ensure inter-organizational linkages across regions [21,33]. Such a strategic approach allows regional actors to promote collaborative scientific and technological projects across Korea, which can gather distant partners and provide opportunities to further develop capabilities in their areas of specialization [25].

### 1.1.2. Organizational Structure of the National Strategy

In Korea, a joint governing body for the implementation of the Korean New Deal was established to channel cooperation and discussion between the political community and the government [11]. This body comprises the heads of the three pillars of the Korean New Deal and relevant ministers, including those for the Ministry of Science and ICT (MSIT), the Ministry of Environment (MOE), and the Ministry of Trade, Industry, and Energy (MOTIE). Subsequently, the Ministry of the Interior and Safety (MOIS) and regional mayors were invited to implement the Regionally Balanced New Deal in cooperation with the central government and 17 local governments. As shown in Figure 1, this governing body provides overarching information about the current status of national R&D activities across the country in order to establish collaborative projects between central and local governments for reducing regional inequality via a national strategy.

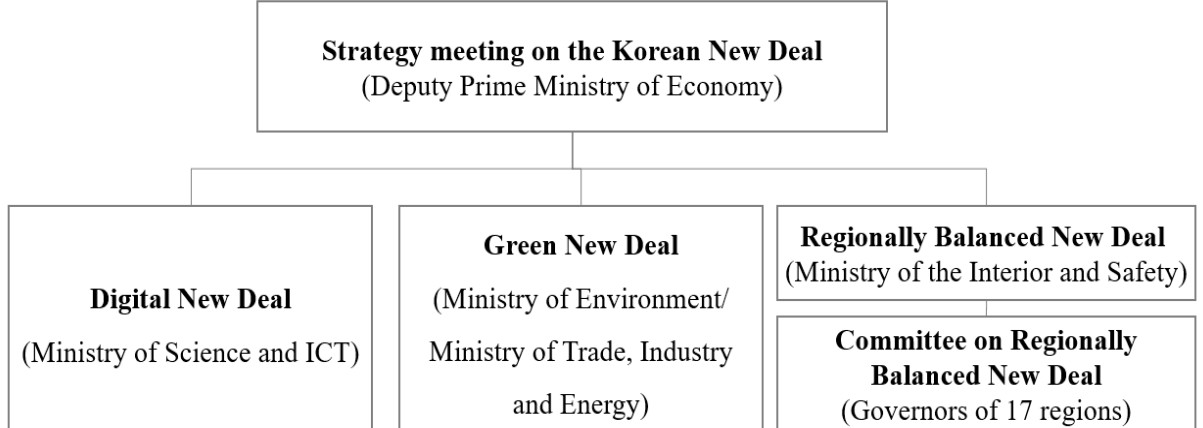

**Figure 1.** Organizational structure of the national strategy encompassing the Korean New Deal, Digital New Deal, Green New Deal, and Regionally Balanced New Deal.

The government also plans to launch a task force for the promotion and spread of the Korean New Deal and hold strategic meetings. The strategic meeting is an important decision-making organization chaired by the president and consists of joint government ministries, the Korea New Deal Committee, local governments, and private companies. The strategic meeting will be organized once or twice a month to discuss various forms of cooperation, including comprehensive reporting, sharing major project progress, private demand and investment, cooperation with local governments, and system improvement [11]. Although the Korean government established the organizational structure of a national strategy, in practice, any systematic framework to discuss the trans-regional collaborative strategy planning is not provided to the members of the task force.

### 1.1.3. Future Mobility Policies of Korea

Emissions from the transport sector are a major contributor to climate change and account for approximately 20% of global $CO_2$ emissions, with road transport accounting for three-quarters of all transport emissions [34]. Moreover, $CO_2$ emissions from road transportation in Korea comprise 14% of all $CO_2$ emissions in Korea [35]. Under the 2015 Paris Agreement, the Korean government developed a carbon-neutral strategy that would both achieve the relevant target and further diversify transportation fuel types, acknowledging that electrified and hydrogen vehicles can decrease oil imports (Korea is the fifth-largest importer worldwide) and utilize electricity and hydrogen from renewable sources produced in Korea [36–38]. Moreover, the Korean government has been strongly invested in supporting the development of new technologies predicted to prevail in future markets.

This is because the automobile industry has a huge influence on the economy in Korea, which is the home of the headquarters and factories of the Hyundai Motor Group (the fifth-largest automobile manufacturing company worldwide) [39,40]. As such, the government implemented the Eco-Friendly Mobility of the Future (hereafter future mobility) project as a key project of the Green New Deal. Under the Green New Deal, the Korean government set up the industrial goal for the future mobility project. More precisely, the project aims to supply 1.13 electric vehicle units, including passenger cars, buses, and trucks by 2025, and install 15,000 quick chargers and 30,000 slow chargers. The project proposes to extend the driving range of electric vehicles from the current 400 km to 600 km by 2025 and shorten the charging speed from 40 min to 15 min. Additionally, 200,000 hydrogen vehicles and 450 charging facilities will be provided by 2025. The project proposes to develop hydrogen cars with 500,000-kilometer durability at a price of KRW 40 million (USD 41,000) by 2025 [11].

Two zero-emission vehicle technologies—battery electric vehicles and fuel cell (electric) vehicles—have emerged as the main pillars of future mobility [41,42]. Although the battery electric vehicle market is currently dominating in several countries such as the US, EU, China, and Korea, industrial and government efforts to spur the market development of fuel cell transport are supported by numerous advantages relative to batteries, including refueling times roughly comparable to gasoline, longer driving ranges, fewer space requirements for hydrogen refueling stations, less performance deterioration from battery aging, and less reliance on lithium and cobalt supply chains [39,42]. Meanwhile, autonomous technologies can improve road safety, increase fuel efficiency and reduce emissions, and improve urban public transportation via multimodal transportation services [43]. Therefore, electric vehicles are likely to integrate autonomous technologies and would play an important role in promoting zero-emission and sustainable transportation.

In this study, we use the future mobility project as a case study to propose an information framework for implementing a collaborative R&D strategy plan between central and local government linking the Regionally Balanced New Deal and Green New Deal in Korea. The absence of a systematic framework, which is the primary limitation of the previous studies [7,8,10,21,27,31,33,41,44], is overcome by this study.

### 1.1.4. Need for Developing a Systematic Framework for a Collaborative Trans-Regional R&D Strategy Plan

From examining the inherent policy and political implications in "Green (New) Deal" initiatives of the world's largest economies, including the United States in February 2019, the European Union in December 2019, and South Korea in 2020, some scholars [7,8,10,21,27,45] asserted that these initiatives (green economic stimulus packages) should be stipulated as a sustainability transition strategy coupled with the climate and financial policies to accomplish the dual goals of climate change mitigation and regional inequality reduction. Consequently, a principle remark (a normative statement) that these green economic stimulus packages should be public long-term transition programs that set clear sustainability targets in green cars, transport systems, charging station network, etc., but where budgets are devolved to enable localities to design initiatives appropriate to their needs in collaborations with local stakeholders, was put forward.

Meanwhile, since the transport sector accounts for 21% of total emissions, and road transport accounts for three-quarters of transport emissions, road transport accounts for 15% of total $CO_2$ emissions [34]. Even though COVID-19 restrictions temporally drove down transportation demands, it is expected to grow across the world in the coming decades as the global population increases, incomes rise, and more people can afford cars, trains, and flights [31]. With the advancement and wide acceptance of electric and hydrogen technologies, decarbonization could be attained within a few decades in many regions, including the European Union, United States, China, and Japan [42]. Thus, many scholars investigated automotive industries and transport sectors in major countries to identify the roles/strategies of governments to facilitate the achievement of the greenhouse gas (GHG) emission reduction targets under the Paris Agreement [31,42,46,47].

In particular, only a few previous studies have focused on regional areas in line with the direction of technology policies for easing greenhouse gas emissions in the Korean transport sector under the Paris Agreement [40] or its Green New Deal [41]. Expanding the scope of the country, some studies in Austria [33] and the United Nations [48] require the role of government in developing green sustainable transportation strategies at the regional level, but existing studies have only asserted for the future policy or political implications through the review of three examples of green regional development initiatives or transportation impacts of transport sector during COVID-19 restrictions.

In order to successfully implement these national industrial strategies/stimulus packages, it is required to be aligned with a systematic (investment) framework that can contribute to a continuous collaboration with a range of stakeholders at multiple policy-making, managerial, and administrative levels as well as the engagement of local implementation actors such as local universities, research institutes, autonomous organizations, and a range of local companies [49]. A consensus is that developing a coherent/R&D strategy (planning) should be built on evidence-based situation analysis on particular sectors and technologies, an institutional base for investment monitoring, and process management for inclusive communication among stakeholders [50].

Previous studies [8,10,21,27,31,33,41,44] asserted only the normative statements that planning a collaborative strategy with a trans-regional perspective is important to successfully accomplish these green innovation initiatives and that the existing strategy lacks an explicit framework. Therefore, it is necessary to build a systematic framework for a fine-tuned trans-regional innovation scheme with regional, technological, and organizational dimensions, thereby identifying the reasons for the gap in a regional variation of innovation capabilities and then suggesting appropriate strategies to bridge the gap. The proposed framework is established on the abovementioned consensus on directions for a better, coherent/R&D strategy (planning) to facilitate collaborations with a range of stakeholders via continuous communication.

### 1.1.5. Changes in Sustainable Urban Mobility Modes in the Post-COVID-19 Era

The COVID-19 pandemic had swift and brutal impacts on the operation of the current transportation infrastructure [47,51]. In many nations, safety concerns, anxiety, and stress levels increased in society regarding using public transport after the beginning of the pandemic [51,52]. In the Republic of Korea, experts expected a mobility modal shift of 94.4% from public transport to personal car and only an expected 5.6% shift to bicycles. Almost half (45.2%) of experts expected a shift to a high-carbon mobility mode. Nonetheless, many people still had to use public transport because they did not have alternative modes. Additionally, a resurgence of the COVID-19 pandemic is expected until 2024 and even further [47]. Moreover, during the COVID-19 pandemic, several people changed their perceptions, favoring the use of sustainable mobility modes to protect the climate [52]. Thus, it is required for policy makers to make use of public funds both for improving public transport infrastructure (i.e., Mobility-as-a-Service (MaaS)) and for supporting the technological advancement of green vehicles (i.e., shared autonomous electric and alternative-fuel vehicles) to make our societies highly sustainable (or safe and trusted) in the era of building back with the Paris Agreement and the Sustainable Development Goals (SDGs). These efforts should concentrate on five primary targets—road safety, energy efficiency, sustainable infrastructure, urban access, and reduced fossil fuel subsidies [46,47,51,53]. The direct impact of the COVID-19 pandemic and demographic changes caused by an aging population must be regarded as a basis for the transition toward green and healthy sustainable transport and for increased investment in public transport to match new requirements [48,52].

### 1.2. Research Purpose and Research Questions

In summary, the framework allows central and local governments to promote the exchange of knowledge and other assets within and beyond regions, using overarching information about the present state of national R&D activities from the perspective of technology areas and regions.

To establish a trans-regional collaborative R&D strategy, one must first identify the current status of the target research fields. Then, to reduce stakeholder uncertainty regarding the information on the different statuses of different target fields of knowledge, one must provide comprehensive evidence on the current status of the target research fields [54]. As remarked in a previous study [55], this procedure is key to ensuring strengthened coordination among stakeholders, which, therefore, improves the quality of the decision-making process related to national R&D strategy planning [56]. Thus, the aim of this study is to provide timely, comprehensive, and useful information on the status of R&D activities related to electric and hydrogen vehicles in 17 regions of Korea through an information analysis framework. Our primary research question was:

Research Question RQ1: How much did the Korean government invest in regional future mobility from the perspective of automotive company locations during the period 2015–2020?

In addition to understanding the status and trends of investment of the Korean government in future mobility-related technology areas, we examined the following research questions:

Research Question RQ2-1: What was the distribution of investment in future mobility-related technology areas in 2015–2020?

Research Question RQ2-2: What trends of investment in technology areas emerged during this period?

To provide the information for comprehending the competitiveness of future mobility-related fields in terms of regions, we investigated the status and trends of investment of the Korean government by asking the following research questions:

Research Question RQ3-1: What was the regional distribution of investment in future mobility-related technology areas?

Research Question RQ3-2: What types of organizations (among academia, industry, and research institutes) have played an important role in future mobility-related technology areas from a regional perspective?

Finally, we looked closely at the detailed research activities in future mobility-related research fields for information about potential partners who may share the knowledge among other stakeholders and asked the following research question:

Research Question RQ4: What organizations related to future mobility-related technology areas may serve as trans-regional collaborative R&D partners from a regional perspective?

The remainder of this article is structured as follows. Following this general introduction, the Materials and Methods section describes the framework and methodology. The Results section presents comparative results of the research profiling and machine learning analyses. Finally, the Discussion and Conclusion sections elaborate on the research contributions, implications for practice, and research limitations and indicate promising research opportunities to pursue in the future.

## 2. Materials and Methods

### 2.1. Data Collection and Preprocessing

In this study, we used data on nationally funded R&D projects collected from the National Science & Technology Information Service (NTIS), which holds information on a total of 1411 national R&D projects in Korea from 2015–2020 and is internally operated by the Korea Institute of Science and Technology Information (KISTI), funded by the MSIT of Korea. The organization name, title, and abstract of each project were translated into English. Under the guidance of the experts from universities, research institutes, and industries, the two authors conducted the full search strategy and data collection together using the following keywords and the combination of their variants during the search query: "e-mobility", "automobility", "green car", "electric vehicle", "electric vehicle with battery", "electric vehicle fuel cell", "fuel cell vehicle", "hydrogen fuel cell vehicle", "hybrid electric vehicle with fuel cell", "plug-in hybrid electric vehicle", "autonomous vehicle", "self-driving car", "driverless car", and "connected autonomous vehicle". The dataset is described in Table 1. In total, we collected data from 1411 nationally funded R&D projects related to future mobility and conducted between 2015 and 2020 based on the search terms in Table 1, then added 700 additional project data items based on the 130 R&D programs to which the collected data belonged. Future mobility experts investigated whether individual projects were associated with future mobility and selected a total data sample of 2111 projects, which aimed to cover the entire dataset as much as possible. However, the dataset from NTIS has two critical problems. One problem is that the addresses of the research organizations that completed the projects were often incorrect; however, it is important to categorize the number of projects in each of the 17 regions to understand the status and trend of government R&D investment for future mobility. The other problem is that the type of organization (e.g., academia, industry, research institutes) was also often incorrect. For example, KISTI, which was established and operated by the government, was designated as academia or industry. Correcting the type of organization is critical for forming a potential collaborative network to support green innovation in the industry. Therefore, the address and type of organization were corroborated or rectified as necessary. After removing projects with missing funding information, we acquired a final data sample of 1963 projects with a total funding amount of USD 1285.4 million (Tables 2 and 3).

**Table 1.** Examples of data on R&D public projects in the Korean R&D database (NTIS).

| Regions | Unique Identification Number (ID) | Organization | Type of Organization | Research Program | Funding (USD) | Project Period | | Project Contents | |
|---|---|---|---|---|---|---|---|---|---|
| | | | | | | Start Date | End Date | Title | Abstract |
| Incheon | 1415143912 | Korea Electronic Material Co. | Industry | Material-parts, equipment for infrastructure | 125,000 | 2015-07-01 | 2016-06-30 | Improve the reliability of environmentally friendly cars through the relay contacts material modification and hetero-junction technology optimization | Arc Chamber Assy improving (1) Co. Korea electronic materials (lead organization) conventional contact material composition and process optimization (alloy composition, sintering conditions, etc.)—the electrical properties of the contacts composition and . . . |
| Chungcheongnam-do | 1415139977 | Korea Motor Research Corp. | Institutes | Development of core technology for transportation system industry | 419,583 | 2015-05-01 | 2016-04-30 | Build cognitive skills development and related supply chain systems with self-vehicle platform | autonomous vehicle platform development—autonomous vehicle platform system architecture development—surround sensors and autonomous mandatory system. identification information based on the mission carried out the vehicle path management technology for—Mission generate specific path requirements analysis—mission-specific . . . |
| Gyeonggi-do | 1415143557 | Korea Electronics Technology Institute Corp. | Institutes | Industrial convergence technology, industrial core technology | 1,330,000 | 2015-12-01 | 2018-11-30 | 360 3D Laser scanning-based low-cost, small-vehicle technology LiDAR system developed by real-time monitoring possible | (1) LIDAR transceiver system design and development—LIDAR specifications for the transmission and reception system and the requirements analysis—Laser signal and developing a light source module . . . . (8) point cloud data Emitter modeling and object recognition technology—point cloud data modeling . . . |
| Seoul | 1415140513 | Seoul National University | Academia | Automobile industry technology development (Green Car) | 1,052,067 | 2015-06-01 | 2020-05-31 | Stack and part design technology for the reduction of the number of stacked fuel cell stack borrowing 400 V | (1) The cell unit used in the study in each of the participating institutions Design and Implementation. (6) large area dual cell design and graphite separation applying the trial product manufactured, performance analysis of Flow stack manifold designs and separated by plate internal manifold designs . . . |

**Table 1.** *Cont.*

| Regions | Unique Identification Number (ID) | Organization | Type of Organization | Research Program | Funding (USD) | Project Period | | Project Contents | |
|---|---|---|---|---|---|---|---|---|---|
| | | | | | | Start Date | End Date | Title | Abstract |
| Jeollabuk-do | 1415140517 | Iljin Composites Company LTD. | Others | Automobile industry technology development (Green Car) | 1,491,667 | 2018-06-01 | 2022-12-28 | Improve the reliability of environmentally friendly cars through the relay contacts material modification and hetero-junction technology optimization | Low Permeation/lightweight liner developed—and developing lightweight liner through hydrogen high barrier material development. High-performance epoxy (Toughened epoxy) Development . . . . the silica using a filler, the impact resistance reinforcing Toughened epoxy—the manufacture composite . . . |

**Table 2.** Future mobility-related nationally funded project data and search terms.

| Search Terms | Time Period | Amount of Raw Data | Number of Data Utilized |
|---|---|---|---|
| ((electric * OR electrified OR battery OR hybrid * OR plug-in * OR hydrogen * OR fuel cell OR autonomous OR green OR driverless OR connected OR self-driving) AND (vehicle * OR car * OR automobile* OR transportation *)) | 2015–2020 | 2111 | 1963 |

Asterisks (*) in search terms were used to broaden the search by finding words that start with the same letters.

**Table 3.** Number of projects nationally funded by research organizations in different regions.

| Region | Funding (USD Million) | No. Projects | Funding Per Project | Funding (%) |
|---|---|---|---|---|
| Gangwon-do | 2.9 | 14 | 2.5 | 0.2% |
| Gyeonggi-do | 372.5 | 504 | 8.9 | 29.0% |
| Gyeongsangnam-do | 27.1 | 35 | 9.3 | 2.1% |
| Gyeongsangbuk-do | 15.3 | 87 | 2.1 | 1.2% |
| Gwangju | 20.2 | 58 | 4.2 | 1.6% |
| Daegu | 64.5 | 92 | 8.4 | 5.0% |
| Daejeon | 196.8 | 283 | 8.3 | 15.3% |
| Busan | 4.5 | 41 | 1.3 | 0.3% |
| Seoul | 207.0 | 394 | 6.3 | 16.1% |
| Sejong | 27.4 | 69 | 4.8 | 2.1% |
| Ulsan | 15.8 | 38 | 5.0 | 1.2% |
| Incheon | 47.9 | 59 | 9.7 | 3.7% |
| Jeollanam-do | 10.1 | 11 | 11.1 | 0.8% |
| Jeollabuk-do | 54.6 | 59 | 11.1 | 4.2% |
| Jeju | 20.8 | 26 | 9.6 | 1.6% |
| Chungcheongnam-do | 148.9 | 131 | 13.6 | 11.6% |
| Chungcheongbuk-do | 49.2 | 62 | 9.5 | 3.8% |
| Total/Average | 1285.4 | 1963 | 7.9 | 100.0% |

To understand the disciplinary characteristics of the research fields related to these nationally funded research projects, we used the All Science Journal Classification (ASJC) model in the machine learning process by employing author keywords from approximately one million articles found in Scopus (i.e., the features) and the ASJC codes assigned to each paper (i.e., the labels). Then, according to the similarity (calculated according to the ASJC classification model) and the procedures of previous studies [54,55], we assigned three ASJC codes to each nationally funded research project. The probability of the relevance of each assigned ASJC code was determined based on the title and abstract of the nationally funded research project. Moreover, we applied the 10% threshold probability to improve our understanding of the correlation between the assigned ASJC codes and the nationally funded research projects. A conceptual diagram of this process is shown in Figure 2.

### 2.2. Co-Occurrence Matrix

Based on previous studies [54,55,57], we used a co-occurrence technique to identify future mobility-related research fields. The number of times ASJC codes appeared together in a project group revealed the relevance of that project. Specifically, the co-occurrence matrix showed the number of times that element *i* (from the first list) and element *j* (from the second list) appeared together in the text; namely, *i,j* = ASJC codes. The more often the ASJC codes appeared in the projects, the higher the relevance of the projects with those ASJC codes.

### 2.3. Clustering and Network Visualization

The network was created based on the number of appearances of ASJC codes in the projects, which was defined by the co-occurrence matrix. All nodes in the network were drawn based on the titles of the research fields present in the ASJC codes, and the font size indicated the frequency of the co-occurrence of each ASJC code in comparison with other codes. By visualizing this network structure, we can understand the relationship between ASJC codes. The mapping and clustering were calculated based on minimizing Equation (1), which describes the clustering algorithm and is explained in a previous study [55]:

$$V(x_1, \ldots, x_{n,}) = \sum_{i<j} \frac{2mc_{ij}}{c_i c_j} d_{ij}^2 - \sum_{i<j} d_{ij} \qquad (1)$$

here, $n$ is the number of nodes in the network, $m$ is the number of links in the network, $c_{ij}$ is the number of links between nodes $i$ and $j$, and $c_i$ is the number of nodes $i$.

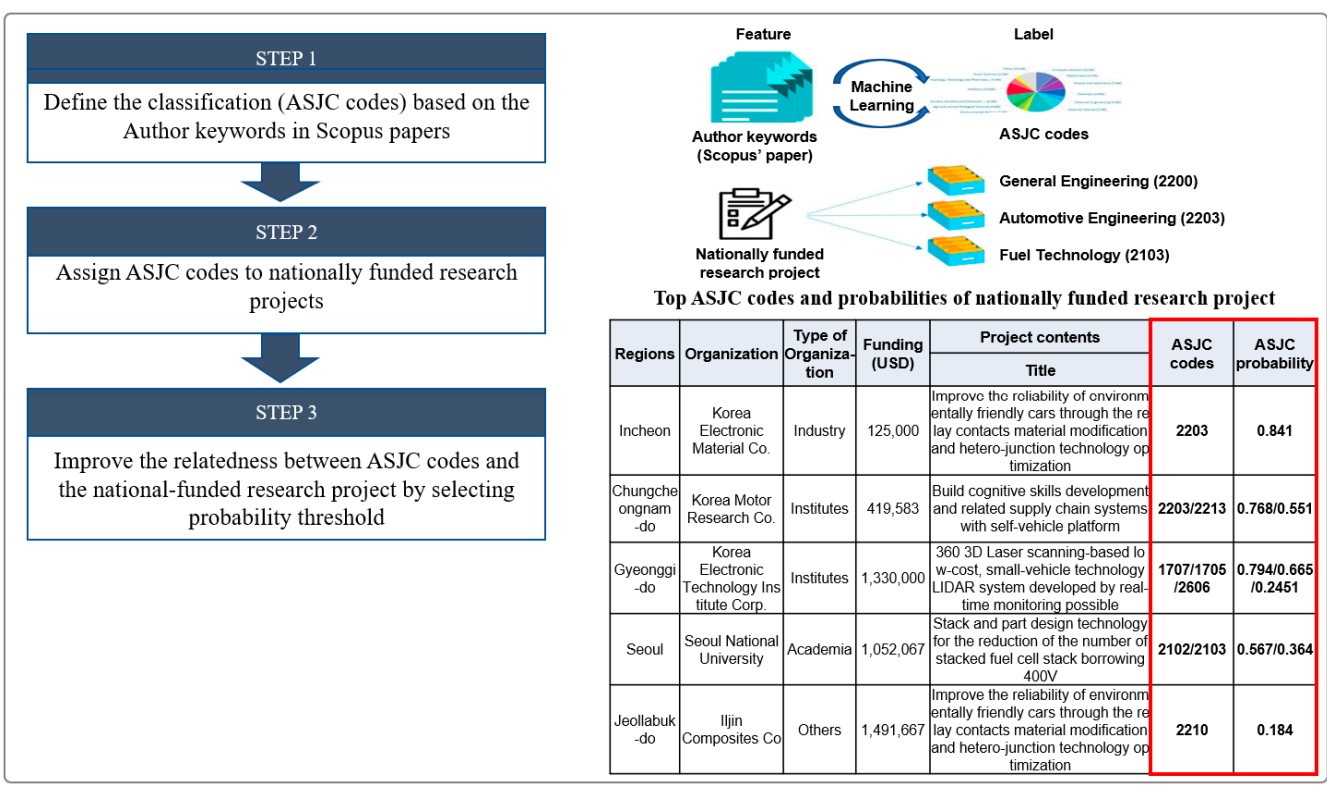

**Figure 2.** Process of assigning ASJC codes to nationally funded research projects and improving the correlation between ASJC codes and projects.

With respect to $x_i, \dots, x_n$; $d_{ij}$ is the distance between nodes $i$ and $j$. For the mapping, $d_{ij}$ was calculated by the following formula:

$$d_{ij} = x_i - x_j = \sqrt{\sum_{k=1}^{p} \left( x_{ik} - x_{jk} \right)^2} \qquad (2)$$

where $x_i$ is a vector denoting the location of node $i$ in a $p$-dimensional map. For the clustering, $d_{ij}$ was calculated by the following formula:

$$d_{ij} = \begin{cases} 0 & if \ x_i = x_j \\ \frac{1}{\gamma} & if \ x_i \neq x_j \end{cases} \qquad (3)$$

where $x_i$ = integer denoting the cluster to which node $i$ belongs, $\gamma$ = resolution parameter.

The resolution parameter ($\gamma > 0$) determines the level of detail of the clustering; the higher the value of the parameter, the larger the number of clusters produced. The number of clusters ranged from 1 ($\gamma = 0.1$) to 7 ($\gamma = 1.0$). To conduct the semantic network analysis, we chose seven clusters according to the number and combination of items (i.e., ASJC codes) in individual clusters. With seven clusters, the fields of the ASJC system were divided into 2700 general engineering fields, and 3100 general physics and astronomy fields, where the former was designated as a tool for safety, and the latter was created as an object recognition-related field, which allowed us to provide strategic insights.

*2.4. Defining the Goals of Future Mobility-Related Research Fields*

The future mobility-related research fields can only be defined by analyzing the title and abstract of the R&D projects and the distribution of ASJC codes. This is supported by discussions between experts on future mobility-related research, who can provide relevant knowledge and expertise for investigating particular research fields. Therefore, we first assessed the approximate research fields by determining the distribution of ASJC codes comprising each cluster. We then identified the title and abstract contents of the R&D project in the clusters. Finally, we determined the goal of the research field of each cluster. The entire process is depicted in Figure 3.

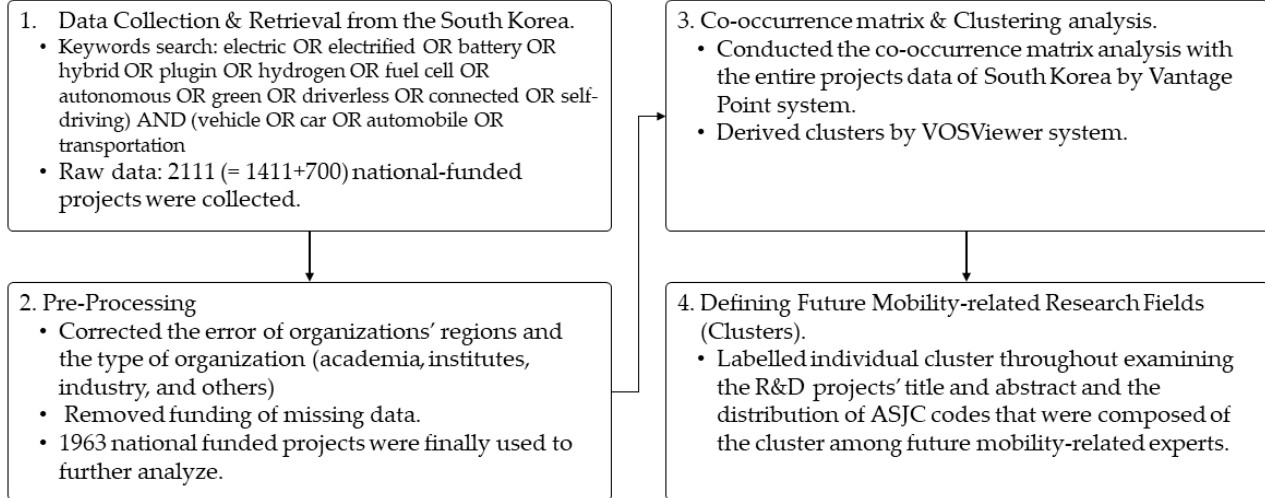

**Figure 3.** Process of data collection and analysis of nationally funded global research projects related to future mobility.

## 3. Results

*3.1. Future Mobility-Related Research Fields of Nationally Funded Projects*

The network visualization of future mobility-related research fields is shown in Figure 4. In this study, the items/nodes, which are treated as the objects of interest, refer to the research fields (i.e., ASJC codes). The links, which imply a relationship between two items, refer to co-occurrence links between research fields. The strength/weight of a link indicates the number of projects in which the two research fields appear together. The label size and circle size for each research field are determined by its weight; namely, the higher the weight of a research field, the larger its label and circle. The characteristics of each research field were determined by the cluster to which it belonged.

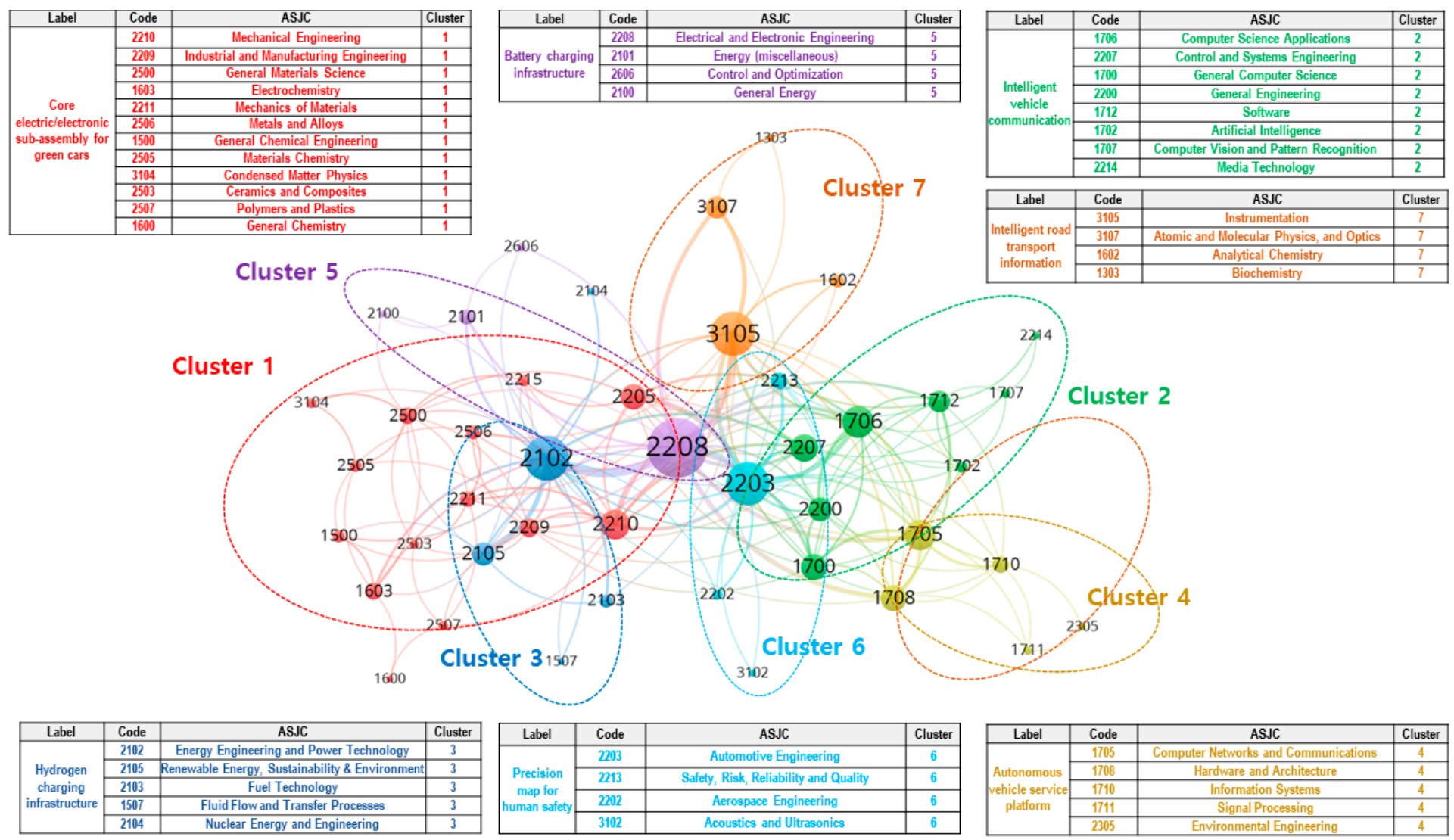

| Label | Code | ASJC | Cluster |
|---|---|---|---|
| | 2210 | Mechanical Engineering | 1 |
| | 2209 | Industrial and Manufacturing Engineering | 1 |
| | 2500 | General Materials Science | 1 |
| | 1603 | Electrochemistry | 1 |
| Core | 2211 | Mechanics of Materials | 1 |
| electric/electronic | 2506 | Metals and Alloys | 1 |
| sub-assembly for | 1500 | General Chemical Engineering | 1 |
| green cars | 2505 | Materials Chemistry | 1 |
| | 3104 | Condensed Matter Physics | 1 |
| | 2503 | Ceramics and Composites | 1 |
| | 2507 | Polymers and Plastics | 1 |
| | 1600 | General Chemistry | 1 |

| Label | Code | ASJC | Cluster |
|---|---|---|---|
| | 2208 | Electrical and Electronic Engineering | 5 |
| Battery charging | 2101 | Energy (miscellaneous) | 5 |
| infrastructure | 2606 | Control and Optimization | 5 |
| | 2100 | General Energy | 5 |

| Label | Code | ASJC | Cluster |
|---|---|---|---|
| | 1706 | Computer Science Applications | 2 |
| | 2207 | Control and Systems Engineering | 2 |
| | 1700 | General Computer Science | 2 |
| Intelligent | 2200 | General Engineering | 2 |
| vehicle | 1712 | Software | 2 |
| communication | 1702 | Artificial Intelligence | 2 |
| | 1707 | Computer Vision and Pattern Recognition | 2 |
| | 2214 | Media Technology | 2 |

| Label | Code | ASJC | Cluster |
|---|---|---|---|
| Intelligent road | 3105 | Instrumentation | 7 |
| transport | 3107 | Atomic and Molecular Physics, and Optics | 7 |
| information | 1602 | Analytical Chemistry | 7 |
| | 1303 | Biochemistry | 7 |

| Label | Code | ASJC | Cluster |
|---|---|---|---|
| | 2102 | Energy Engineering and Power Technology | 3 |
| Hydrogen | 2105 | Renewable Energy, Sustainability & Environment | 3 |
| charging | 2103 | Fuel Technology | 3 |
| infrastructure | 1507 | Fluid Flow and Transfer Processes | 3 |
| | 2104 | Nuclear Energy and Engineering | 3 |

| Label | Code | ASJC | Cluster |
|---|---|---|---|
| | 2203 | Automotive Engineering | 6 |
| Precision | 2213 | Safety, Risk, Reliability and Quality | 6 |
| map for | 2202 | Aerospace Engineering | 6 |
| human safety | 3102 | Acoustics and Ultrasonics | 6 |

| Label | Code | ASJC | Cluster |
|---|---|---|---|
| | 1705 | Computer Networks and Communications | 4 |
| Autonomous | 1708 | Hardware and Architecture | 4 |
| vehicle service | 1710 | Information Systems | 4 |
| platform | 1711 | Signal Processing | 4 |
| | 2305 | Environmental Engineering | 4 |

**Figure 4.** Future mobility-related research fields.

The future mobility-related research fields were divided into seven clusters. After considering the titles and abstracts of the projects, their representative research fields, and the associated keywords, we created the ultimate goals pursued by each research field, which are described as follows:

- Cluster 1. Core electric/electronic sub-assembly for green cars. Research on the core electric and electronic sub assembly for autonomous and electric vehicles; dynamic control applications to future cars (i.e., electric motor, power conversion system).
- Cluster 2. Intelligent vehicle communication. Research on vehicle-to-vehicle and vehicle-to-infrastructure communication for controlling road transport service (i.e., V2X: Vehicle-to-everything, AUTOSAR Adaptive Platform).
- Cluster 3. Hydrogen charging infrastructure. Research on production and storage for hydrogen vehicles, charging interfaces, and management of hydrogen stations.
- Cluster 4. Autonomous vehicle service platform. Research on vehicle platforms, edge platforms, and clouds for autonomous driving services (i.e., MaaS: Mobility-as-a-Service, TaaS: Transportation as a Service).
- Cluster 5. Battery charging infrastructure. Research on production and storage for electric vehicles, charging interfaces, and management of electric stations.
- Cluster 6. Precision maps for human safety. Research on dynamic and static map generation, high precision positioning, and human-machine interface for autonomous vehicles (i.e., LDM: Local Dynamic Map, IMU: Inertial Measurement Unit).
- Cluster 7. Intelligent road transport information. Research on road and environment conditions, including object detection and tracking, pedestrian detection, and road sign recognition-based optimal driving support services for autonomous vehicles (i.e., C-ACC: Cooperative Adaptive Cruise Control, LCS: Lane Control System).

The seven clusters reflected the technological directions of the automotive industry that electric vehicles have integrated into the widespread advancement and adoption of alternative fuels, autonomous vehicles, and MaaS [58]. Electric vehicles that draw electricity stored in rechargeable battery packs or a fuel cell powered by hydrogen to power electric motors with motor controllers were classified into Cluster 1. In order to reduce carbon emissions and air pollution; operate at optimal efficiency; increase safety for drivers, passengers, as well as for bicyclists and pedestrians; and enhance the traffic flows and multimodality to decrease urban traffic congestion (Cluster 6 and Cluster 7), many governments plan to expand the electrification of road transport by providing battery charging/hydrogen refueling infrastructure (Cluster 5/Cluster 3) and connectivity (i.e., V2X) infrastructure (Cluster 2) and accelerating the eco-friendly car commercialization with spurring R&D activities such as sustainable mobility services for urban transport with MaaS and/or TaaS (Cluster 4). In the following subsections, we present the status or trend of nationally funded projects for future mobility in Korea in terms of technology clusters, regions, and organizations.

### 3.2. Status of Government Investment in Future Mobility

3.2.1. Status of Government-Funded Projects According to Regions

The government invested USD 1285.4 million in future mobility during 2015–2020. As shown in Figure 5, the CR comprised the largest share of R&D investment of 48.8% (USD 627.4 million), which was divided between the jurisdictions of Seoul (16.1%; USD 207 million), Incheon (3.7%; USD 48 million), and Gyeonggi Province(do) (29.0%; USD 372 million). After the CR, 26.6% of total national research funding for future mobility was invested in Daejeon (15.3%; USD 197 million) and Chungcheongnam-do (11.6%; USD 149 million). Daegu and Jeollabuk-do received 5.0% (USD 65 million) and 4.2% (USD 55 million), respectively. The relationship between the investment status and the automobile industry in each region was then investigated. The information about the investment proportion of future mobility-related research in the 17 regions of Korea may enable a range of stakeholders to provide opportunities for discussions regarding an appropriate R&D investment for the improvement of these regions.

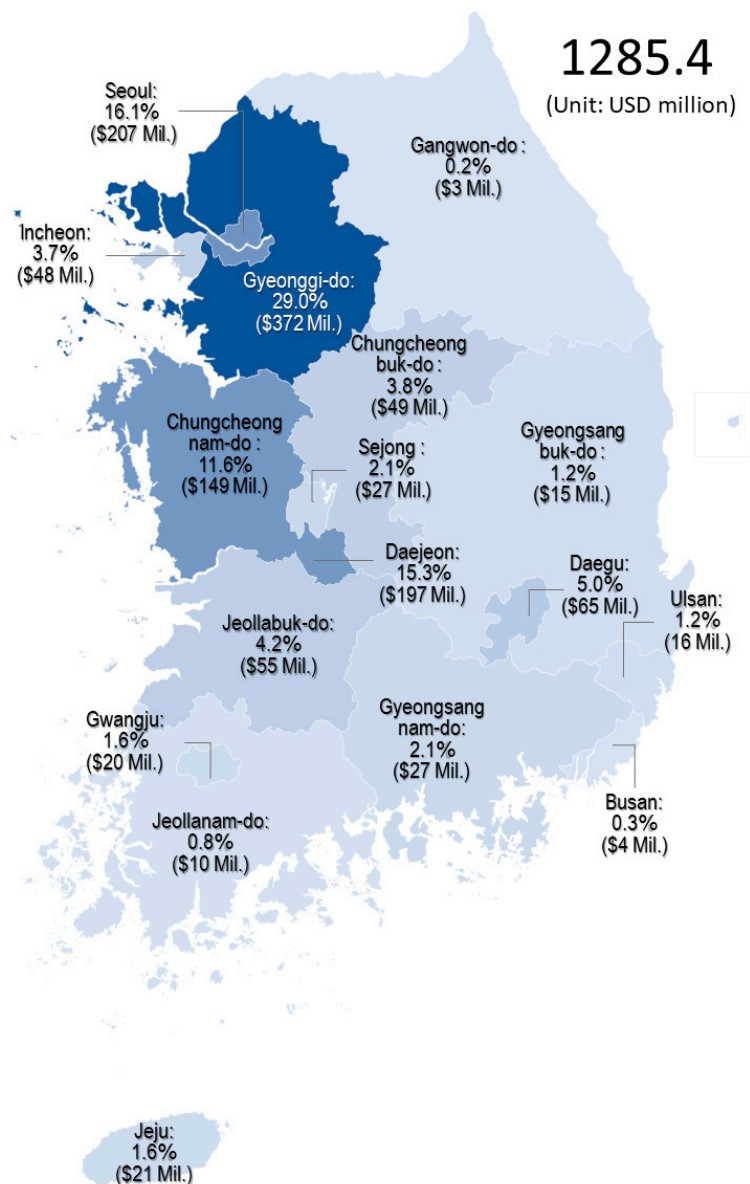

**Figure 5.** Proportion of future mobility-related research in the 17 regions of Korea.

The Hyundai Motor Group, formed through the purchase of Korea's second-largest car manufacturer, KIA Motors, in 1998, is ranked at the fifth and fourth places in the global light vehicle and electric vehicle markets, with sales at 6.69 million and 60,000 units, respectively [59]. The Hyundai Motor Group Headquarters is in Seoul, the Hyundai-Kia Motors R&D Center, KIA Sohari Plant, and KIA Hwaseong Plant are in Gyeonggi-do, the Hyundai Asan Plant and KIA Seosan Plant are in Chungcheongnam-do, Hyundai plants and located in Jeollabuk-do and Ulsan, and a KIA plant is in Gwangju. Therefore, nationally funded research projects are concentrated in the locations of the Hyundai Motor Group's headquarters, R&D centers, and plants. Daejeon and Daegu are the exceptions to this rule. Daejeon has 18 universities and the Daedeok Innopolis (Daedeok Research and Development Special Zone), which is composed of 28 government-funded research institutions, as well as 79 private research institutes with as many as 20,000 researchers, enabling the region to conduct considerable research and development activities. Daegu is the third-largest city in Korea, with a population of over 2.4 million and nine universities; thus, it is larger than Ulsan (the seventh-largest city with a population of over 1.1 million and two universities). This has led to more research being conducted in Daegu.

3.2.2. Status and Trend of Government-Funded Projects According to Technology Clusters

Figure 6 shows the total amount of national research funding for future mobility in terms of technology clusters. A large proportion of the total national research funding was given to Cluster 2 (intelligent vehicle communication, USD 334 million, 26%), followed by Cluster 1 (core electric/electronic sub-assembly for green cars, USD 245 million, 19%). Small amounts of funding were invested in Cluster 4 (autonomous vehicle service platform, USD 88 million, 7%) and Cluster 3 (hydrogen charging infrastructure, USD 100 million, 8%). Because Cluster 1 and 2 are considered the core technology areas for improving the performance of future mobility, considerable investment in these areas is expected [31,60]. It is important for stakeholders to determine the optimal portfolio of R&D projects to maximize the long-term development strategy by comparing heterogeneous R&D projects that are interrelated and aligned with strategies [61]. Hence, it is prerequired to classify projects for facilitating the process of prioritizing R&D projects [62]. This study presents the systematic analysis procedure to classify future mobility-related national projects into technology groups and then provides information about the investment trends of individual technology groups (clusters) to the stakeholders. Thus, the framework opens the starting point to discuss the future of the R&D portfolio to facilitate communication and ensure understanding among them.

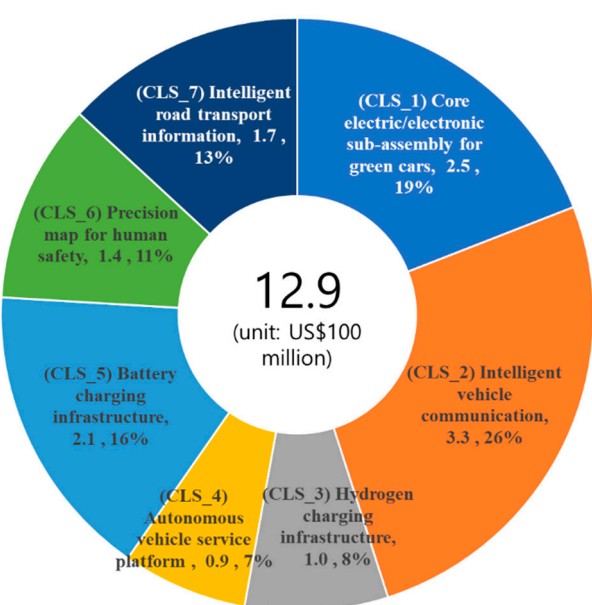

**Figure 6.** Scale of national funding by technology cluster.

In consideration of the global COVID-19 pandemic, we excluded the year 2020 when calculating the compounded annual growth rate, as shown in Table 4. The importance of technology has already been emphasized; thus, Cluster 1 (core electric/electronic sub-assembly for green cars) grew the second-fastest, at a compound annual rate of 65.7%, to become the second-largest cluster, ballooning from USD 9.8 million in 2015 to USD 74.2 million in 2019. Moreover, Cluster 2 (intelligent vehicle communication) continued to be the largest, growing at a compound growth rate of 64.0% from USD 11.6 million to USD 83.7 million, whereas Cluster 5 (battery charging infrastructure) steadily grew at a compound growth rate of 31.7% from USD 17.2 billion to USD 51.6 billion. Cluster 4 (autonomous vehicle service platform) continued to exhibit the fastest growth, increasing with a compound growth rate of 116.3% from USD 1.1 million to USD 24.9 million, and Cluster 7 (intelligent road transport information) continued to exhibit the third-fastest growth, with a compound growth rate of 34.0% and growth from USD 12.3 million to USD 39.7 million. These findings reflect the government's goal to prepare for the expansion of the domestic electric and hydrogen vehicle market. Conversely, Cluster 6 (precision

maps for human safety) increased at the lowest compound growth rate of 15.5% from USD 12.0 million to USD 21.4 million, which may indicate that this technology area is in a mature stage. Meanwhile, Cluster 3 (hydrogen charging infrastructure) grew at a compound growth rate of 51.9% from USD 5.1 billion to USD 27.1 billion, which implies that Korea is focusing on developing a hydrogen economy by increasing the production and use of hydrogen vehicles and establishing an ecosystem for the production and distribution of hydrogen and related technologies [34].

**Table 4.** Temporal trends of national funding for different technology clusters.

| Funding (USD Million) | 2015 | 2016 | 2017 | 2018 | 2019 | 2020 | Total | 15–19 CAGR |
|---|---|---|---|---|---|---|---|---|
| (CLS_1) Core electric/electronic sub-assembly for green cars | 9.8 | 25.4 | 47.8 | 51.9 | 74.2 | 36.2 | 245.4 | 65.7% |
| (CLS_2) Intelligent vehicle communication | 11.6 | 43.4 | 63.4 | 69.3 | 83.7 | 62.3 | 333.7 | 64.0% |
| (CLS_3) Hydrogen charging infrastructure | 5.1 | 10.5 | 16.4 | 25.3 | 27.1 | 16.1 | 100.4 | 51.9% |
| (CLS_4) Autonomous vehicle service platform | 1.1 | 6.0 | 10.8 | 20.0 | 24.9 | 25.0 | 87.8 | 116.3% |
| (CLS_5) Battery charging infrastructure | 17.2 | 22.4 | 40.2 | 52.0 | 51.6 | 25.5 | 208.8 | 31.7% |
| (CLS_6) Precision map for human safety | 12.0 | 24.2 | 32.8 | 32.4 | 21.4 | 17.3 | 140.3 | 15.5% |
| (CLS_7) Intelligent road transport information | 12.3 | 18.2 | 39.4 | 32.2 | 39.7 | 27.2 | 169.0 | 34.0% |
| Total | 69.2 | 150.0 | 250.9 | 283.2 | 322.7 | 209.5 | 1285.4 | 47.0% |

### 3.2.3. Status of Government-Funded Projects According to Technology Clusters and Regions

We also investigated the status of nationally funded projects according to technology clusters and regions to identify the strength of regional technological competitiveness, as shown in Table 5. As shown previously, Korean research capacities related to future mobility are concentrated in Seoul, Gyeonggi-do, and Daejeon. In more detail, Gyeonggi-do received the highest investment in the intelligent vehicle communication cluster (USD 140.7 million), followed by Seoul (USD 83.2 million) and Daejeon (USD 44.4 million), with Chungcheongnam-do (USD 22.6 million) in fourth place. For the autonomous vehicle service platform cluster, Gyeonggi-do and Daejeon came in first and second place with USD 30.6 million and USD 18.9 million, respectively, whereas Daejeon (USD 45.7 million), Gyeonggi-do (USD 35.8 million), and Seoul (USD 27.5 million) received the most investment in the precision maps for human safety cluster. However, some regions exhibited relative advantages or the potential for growth in certain technological domains. Specifically, Chungcheongnam-do has developed relative competitiveness in the core electric/electronic sub-assembly for green cars cluster, with USD 57.2 million of investment, followed by Gyeonggi-do (USD 61.7 million), Incheon (USD 28.6 million), and Seoul (USD 23.9 million). Moreover, Chungcheongnam-do and Jeollabuk-do specialized in the hydrogen charging infrastructure cluster, with USD 43.6 million and USD 31.2 million of investment, respectively. In addition, some regions such as Jeju, Jeollabuk-do, and Daegu placed a research focus on specific technological domains. For the battery charging infrastructure cluster, Gyeonggi-do received the most investment of USD 65.3 million, followed by Seoul (USD 20.4 million), Jeju (USD 20.0 million), and Jeollabuk-do (USD 18.9 million). In the intelligent road transport information cluster, Daejeon received the most investment of USD 56.3 million, followed by Seoul (USD 42.9 million), Gyeonggi-do (USD 26.1 million), and Daegu (USD 25.8 million). The status map of the 17 regions of Korea by seven future mobility-related research fields is illustrated in Figure 7.

**Table 5.** Status of future mobility-related research fields in the 17 regions of Korea.

| (Unit: USD Million) | Total | (CLS_1) Core Electric/ Electronic Sub-Assembly for Green Cars | (CLS_2) Intelligent Vehicle Communication | (CLS_3) Hydrogen Charging Infrastructure | (CLS_4) Autonomous Vehicle Service Platform | (CLS_5) Battery Charging Infrastructure | (CLS_6) Precision Map for Human Safety | (CLS_7) Intelligent Road Transport Information |
|---|---|---|---|---|---|---|---|---|
| Gangwon-do | 2.9 | 0.4 | 0.5 | - | - | 1.9 | - | - |
| Gyeonggi-do | 372.5 | 61.7 | 140.7 | 12.2 | 30.6 | 65.3 | 35.8 | 26.1 |
| Gyeongsangnam-do | 27.1 | 8.1 | 0.0 | 3.6 | - | 15.4 | - | - |
| Gyeongsangbuk-do | 15.3 | 2.3 | 0.8 | - | 0.3 | 5.5 | 2.4 | 4.0 |
| Gwangju | 20.2 | 3.1 | 2.0 | 0.4 | - | 3.9 | 3.4 | 7.3 |
| Daegu | 64.5 | 9.0 | 7.1 | - | 5.8 | 13.6 | 3.2 | 25.8 |
| Daejeon | 196.8 | 16.0 | 44.4 | 4.6 | 18.9 | 10.8 | 45.7 | 56.3 |
| Busan | 4.5 | 2.3 | 0.2 | 0.9 | 0.0 | 0.8 | - | 0.3 |
| Seoul | 207.0 | 23.9 | 83.2 | 1.6 | 7.6 | 20.4 | 27.5 | 42.9 |
| Sejong | 27.4 | 0.5 | 18.0 | - | 6.9 | 1.6 | 0.1 | 0.3 |
| Ulsan | 15.8 | 14.1 | 0.3 | 0.1 | 0.0 | 0.2 | 0.9 | 0.3 |
| Incheon | 47.9 | 28.6 | 2.6 | 2.1 | 0.2 | 0.0 | 12.8 | 1.6 |
| Jeollanam-do | 10.1 | 1.6 | - | - | 0.2 | 8.3 | - | - |
| Jeollabuk-do | 54.6 | 3.7 | - | 31.2 | 0.6 | 18.9 | - | 0.2 |
| Jeju | 20.8 | - | 0.1 | - | 0.8 | 20.0 | - | - |
| Chungcheongnam-do | 148.9 | 57.2 | 22.6 | 43.6 | 14.6 | 6.4 | 3.2 | 1.3 |
| Chungcheongbuk-do | 49.2 | 13.0 | 11.2 | - | 1.1 | 15.5 | 5.3 | 2.9 |
| Total | 1285.4 | 245.4 | 333.7 | 100.4 | 87.8 | 208.8 | 140.3 | 169.0 |

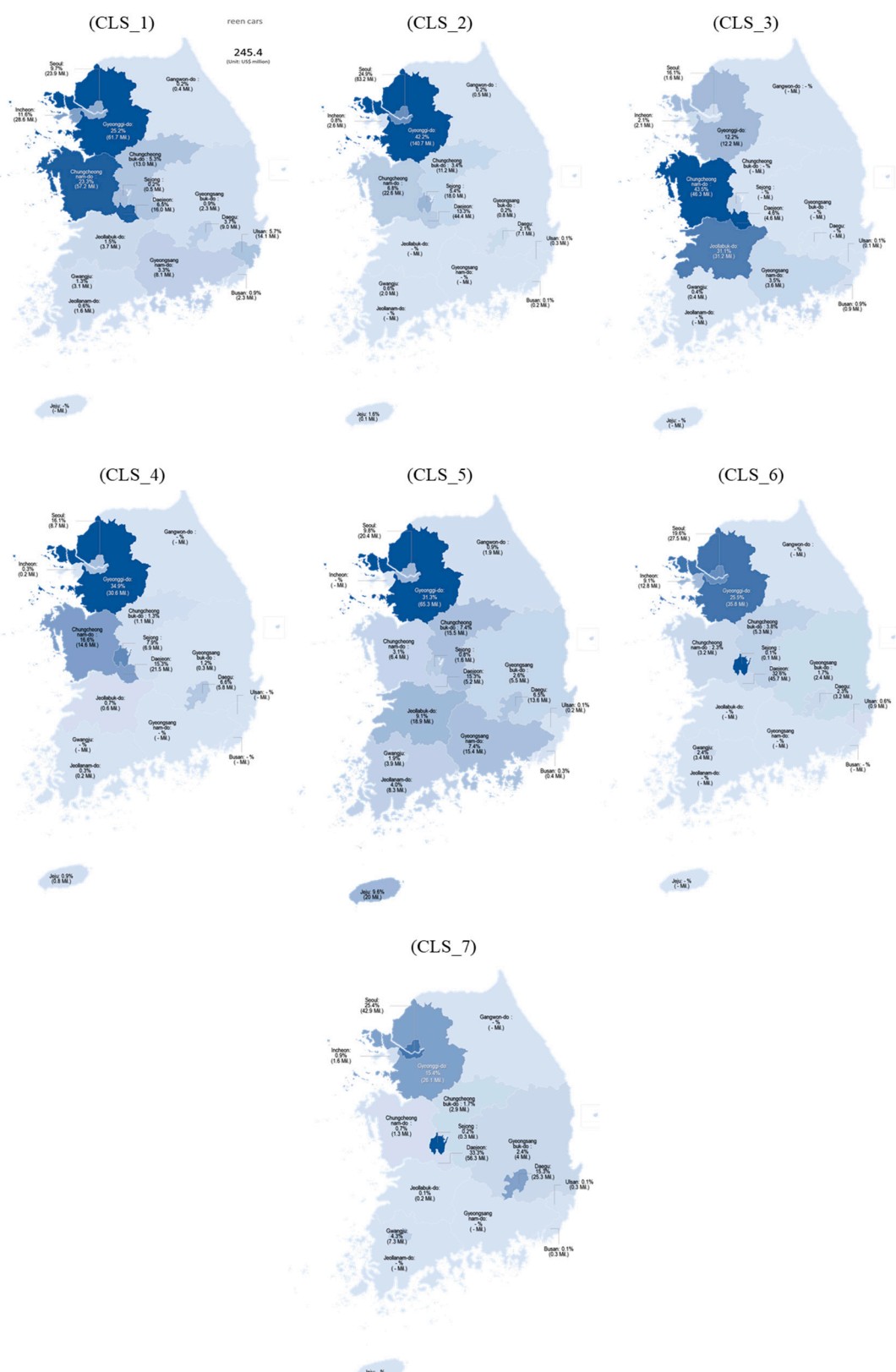

**Figure 7.** Status maps of the 17 regions of Korea by 7 future mobility-related research fields.

3.2.4. Status of Government-Funded Projects According to Technology Clusters, Regions, and Organization Types

Next, we investigated the status of investment according to technology clusters, regions, and organization type to establish the potential collaborative network between academia, industry, and research institutes in the national future mobility industry (Table 6). Individual regional variations are caused by differences depending on their growth path [28]. Thus, it is hard for regions with limited skills and assets in technology and organizations to overcome such limitations [13]. In order to effectively reduce the difficult conditions, it is necessary to understand which types of organizations play a leading role in creating knowledge in specific research fields. It may become the regional base of collaboration with private companies in regions with less-favored research and innovation systems. This study allows stakeholders to consider regional research strategies that re-create networks of industry, universities, and research institutes in low-innovative regions.

Table 6 is a regional R&D portfolio showing how much investment there has been in R&D areas based on technology represented by each of the seven clusters from the perspective of 17 regions in Korea, indicating where each area is leading with high competitiveness. In addition to the technical area and the regional perspectives, it is subdivided by an R&D innovation organization. These results show the types of competitive organizations leading technological development related to specific clusters in each region and the characteristics of regional R&D. As for the specific funding size and investment ranking, industry was the greatest source of investment in Gyeonggi-do (USD 245.7 million), followed by Seoul (USD 124.8 million) and Incheon (USD 47.2 million). Research institutes represented the largest funders in Daejeon (USD 125.1 million) and Chungcheongnam-do (USD 120.2 million). In more detail, in the core electric/electronic sub-assembly for green cars cluster, industry took the lead in Gyeonggi-do (USD 47.3 million), followed by Incheon (USD 28.2 million) and Seoul (USD 17.9 million). Institutes were the biggest funders of this cluster in Chungcheongnam-do (USD 52.6 million, i.e., KATECH—Korea Automotive Technology Institute), and other organizations were the biggest funders in Ulsan (USD 12.1 million, i.e., Ulsan Technopark, the institution of the regional innovative network). In the intelligent vehicle communication cluster, industry took the lead in Gyeonggi-do (USD 80.4 million), whereas institutes were the biggest funders in Daejeon (USD 35.1 million). Interestingly, academia was the biggest funder in Chungcheongbuk-do (USD 11.2 million). In the hydrogen charging infrastructure cluster, other organizations (i.e., KATECH) were the biggest funders in Chungcheongnam-do, whereas industry was the biggest funder in Jeollabuk-do, with USD 39.7 million and USD 29.1 million, respectively. In the autonomous vehicle service platform cluster, academia was the biggest funder in Daejeon (USD 8.5 million), Seoul (USD 3.5 million), and Daegu (USD 2.2 million). Industry (USD 12.1 million) and other organizations, including the Korea Transportation Safety Authority and Korea Electronics Technology Institute (USD 12.1 million), played an important role in Gyeonggi-do. In the battery charging infrastructure cluster, industry was the biggest funder in Gyeonggi-do (USD 53.1 million), followed by Seoul (USD 16.0 million) and Chungcheongbuk-do (USD 15.5 million). Institutes were the biggest funders in Gyeongsangnam-do (USD 10.4 million) and Daejeon (USD 8.0 million). In the precision maps for human safety cluster, industry took the lead in Gyeonggi-do with USD 25.5 million, followed by Seoul (USD 19.8 million) and Incheon (USD 12.7 million). Notably, institutes represented a significant investment in Daejeon (USD 38.8 million). In the intelligent road transport information cluster, all organizations in Daejeon hold important positions with USD 56.3 million of investment. Other organizations (i.e., the Korea Intelligent Automotive Parts Promotion Institute) played an important role in investment in Daegu (USD 20.2 million).

### 3.2.5. Potential National Collaborative Research Organizations According to Technology Clusters

Many nations have launched multiple initiatives and endeavored to build R&D collaboration strategies for dealing with regional inequality. Such endeavors require policy makers or national R&D program directors to have detailed information in order to create a task force for policy issues or promote trans-regional collaborative R&D.

The central government of Korea significantly increased some incentives for private R&D organizations in low-innovative regions to participate in national R&D projects by establishing knowledge links through industry–academia–institutes collaboration [28]. For companies in low-innovative regions that look for a potential partner, the government must provide detailed information about what organizations carried out what R&D projects in a specific technology field. In addition, the pre-condition for developing a coherent/R&D strategy (planning) is to establish a process for inclusive communication among stakeholders [50]. For private R&D organizations in low-innovative regions, this study provides detailed information about the potential partners in a specific research field. For the central-local government, this study presents information about eligible experts in academia, research institutes, and industry who may participate in an ongoing communication process.

Here, we used the example of Cluster 7 (intelligent road transport information) to present the useful information obtained in this study for trans-regional collaboration. Information on the representative research organization, project title, and amount of funding from academia, research institutes, industry, and other organizations for each of the 17 regions in Korea is shown in Table 7. For example, in the Gyeonggi-do region, the potential collaborative partners include Saetbyul Co., Ltd., Sungkyunkwan University, Korea Electronics Technology Institute (KETI), and Korea Testing Laboratory.

**Table 6.** Representative research organization, project title, and amount of funding from academia, research institutes, industry, and other organizations.

| (Unit: USD Thousand) | Organization | Gangwon-do | Gyeonggi-do | Gyeongsangnam-do | Gyeongsangbuk-do | Gwangju | Daegu | Daejeon | Busan | Seoul | Sejong | Ulsan | Incheon | Jeollanam-do | Jeollabuk-do | Jeju | Chungcheongnam-do | Chungcheongbuk-do | Avg(w/o zero) |
|---|---|---|---|---|---|---|---|---|---|---|---|---|---|---|---|---|---|---|---|
| (CLS_1) Core electric/electronic sub-assembly for green cars | Academia | 308 | 1902 | - | 271 | 142 | 88 | 1339 | 906 | 5956 | 466 | 1216 | 312 | - | 449 | - | 100 | 76 | 966.4 |
| | Institutes | - | 4394 | - | - | 18 | - | 8976 | - | 38 | - | - | - | - | - | - | 52,640 | - | 13,213.1 |
| | Industry | - | 47,260 | 3455 | 1224 | 2961 | 8894 | 5710 | 1349 | 17,859 | - | 741 | 28,244 | 1587 | 3174 | - | 4439 | 12,953 | 9989.3 |
| | Others | 125 | 8187 | 4656 | 821 | - | 22 | - | - | - | - | 12,096 | - | - | 27 | - | - | - | 3705.0 |
| (CLS_2) Intelligent vehicle communication | Academia | - | 1965 | - | 777 | 963 | 2358 | 1147 | - | 40,659 | 500 | 224 | - | - | - | - | - | 11,230 | 6646.9 |
| | Institutes | - | 20,459 | 38 | - | - | - | 35,150 | - | 83 | 11,462 | - | - | - | - | - | 22,573 | - | 14,960.7 |
| | Industry | - | 80,372 | - | - | 1056 | 4753 | 8076 | 187 | 41,085 | 6064 | 80 | 2595 | - | - | 56 | 68 | - | 13,126.5 |
| | Others | 503 | 37,892 | - | - | - | - | - | - | 1346 | - | - | - | - | - | - | - | - | 13,246.8 |
| (CLS_3) Hydrogen charging infrastructure | Academia | - | - | - | - | 417 | 13 | 1284 | 42 | 586 | - | 83 | - | - | - | - | - | - | 403.9 |
| | Institutes | - | - | - | - | - | - | 3263 | - | 38 | - | - | - | - | - | - | 22,508 | - | 8602.9 |
| | Industry | - | 11,828 | 3552 | - | - | - | 28 | 890 | 992 | - | - | 2088 | - | 29,083 | - | 3885 | - | 6543.3 |
| | Others | - | 413 | - | - | - | - | - | - | - | - | - | - | - | 2160 | - | 17,257 | - | 6610.1 |
| (CLS_4) Autonomous vehicle service platform | Academia | - | - | - | 143 | - | 2233 | 8470 | - | 3471 | 135 | 31 | 235 | - | 58 | - | 13 | - | 1643.3 |
| | Institutes | - | 10,554 | - | - | - | - | 1990 | - | - | 4688 | - | - | - | - | - | 14,539 | - | 7942.8 |
| | Industry | - | 12,075 | - | 149 | - | 117 | 8459 | 31 | 4154 | 2102 | - | - | 223 | 577 | 796 | - | 1136 | 2710.8 |
| | Others | - | 7984 | - | - | - | 3453 | - | - | - | - | - | - | - | - | - | - | - | 5718.3 |
| (CLS_5) Battery charging infrastructure | Academia | - | 1385 | 31 | 1398 | 115 | 121 | 980 | 79 | 2946 | - | 200 | 35 | - | 325 | - | 9 | - | 635.3 |
| | Institutes | - | 10,593 | 10,401 | - | - | - | 7992 | - | 1404 | - | - | - | - | 3682 | - | - | - | 6814.6 |
| | Industry | 1932 | 53,098 | 4988 | 4097 | 3826 | 13,455 | 1866 | 717 | 16,046 | 1625 | - | - | 8338 | 6474 | 4308 | - | 15,543 | 9736.6 |
| | Others | - | 273 | - | 3 | - | - | - | - | - | - | - | - | - | 12,133 | 15,662 | 2749 | - | 6164.1 |
| (CLS_6) Precision map for human safety | Academia | - | 3319 | - | 633 | - | 1106 | 753 | - | 7613 | 83 | - | 42 | - | - | - | - | 42 | 1698.9 |
| | Institutes | - | 4851 | - | - | - | - | 38,802 | - | - | - | - | - | - | - | - | 3173 | 5299 | 15,608.6 |
| | Industry | - | 25,471 | - | 1764 | 3399 | 2117 | 6192 | - | 19,821 | - | 859 | 12,743 | - | - | - | - | - | 8629.6 |
| | Others | - | 2114 | - | - | - | - | 65 | - | - | - | - | - | - | - | - | - | - | 1089.6 |
| (CLS_7) Intelligent road transport information | Academia | - | 1350 | - | 1398 | 5891 | 1042 | 5815 | - | 13,509 | 266 | 251 | - | - | 167 | - | 42 | - | 2973.1 |
| | Institutes | - | 9058 | - | - | - | - | 28,951 | - | 4498 | - | - | - | - | - | - | 1112 | - | 10,904.9 |
| | Industry | - | 15,604 | - | 2611 | 1373 | 4557 | 9973 | 251 | 24,857 | - | - | 1580 | - | - | - | 97 | 2933 | 6383.6 |
| | Others | - | 79 | - | - | - | 20,212 | 11,540 | - | - | - | - | - | - | - | - | - | - | 10,610.1 |
| TOTAL | Academia | 308 | 9922 | 31 | 4620 | 7526 | 6960 | 19,787 | 1027 | 74,739 | 1451 | 2004 | 624 | - | 999 | - | 164 | 11,348 | 9434.0 |
| | Institutes | - | 59,909 | 10,439 | - | 18 | - | 125,124 | - | 6061 | 16,149 | - | - | - | - | - | 120,228 | - | 48,275.5 |
| | Industry | 1932 | 245,709 | 11,996 | 9844 | 12,614 | 33,893 | 40,305 | 3425 | 124,814 | 9790 | 1681 | 47,250 | 10,148 | 39,308 | 5159 | 8488 | 37,865 | 37,895.3 |
| | Others | 628 | 56,942 | 4656 | 825 | - | 23,686 | 11,540 | - | 1411 | - | - | 12,096 | - | 14,321 | 15,662 | 20,005 | - | 14,706.6 |

**Table 7.** Representative research organization, project title, and amount of funding from academia, research institutes, industry, and other organizations for each of the 17 regions of Korea.

| Region | Type of Organization | Organization | R&D Title | Project Manager | Funding (USD Thousand) |
|---|---|---|---|---|---|
| Gyeonggi-do | Industry | Saetbyul Co., Ltd. | Development of 75 m, 450 g, 360 degrees 3D real-time imaging LIDAR sensor based on MEMS-pinhole single-cell transceiver for application to unmanned automotive vehicles | Wookyung Shin | 1231.4 |
| | Academia | Sungkyunkwan University | The development of a new stereo matching technology with using different kinds of cameras to acquire distance information | Jaewook Jeon | 397.7 |
| | Institutes | Korea Electronics Technology Institute (KETI) | Development of Low-cost and Small LIDAR System Technology based on 3D Laser scanning for 360° Real-Time Monitoring | Yeongook Moon | 3972.9 |
| | Others | Korea Testing Laboratory | Test Environment and Test Cases development of Image-Based Lane Departure Warning System for Smart Digital Tachograph | Gukju Iim | 79.2 |
| Gyeongsangbuk-do | Industry | Tyco AMP Co., Ltd. | Development of MOD System based on Automotive LWIR Thermal Stereo typed Camera for applying AEB | Sangjun Jung | 2611.1 |
| | Academia | Yeungnam University | MIMO Scanning LIDAR using DS-OCDMA | Yongwan Park | 555.0 |
| Gwangju | Industry | SOS Lab Co., Ltd. | 150 m class for autonomous navigation/ADAS LA of the vehicle is small development | Jisung Jung | 551.5 |
| | Academia | Gwangju Institute of Science and Technology (GIST) | Development of an open dataset and cognitive processing technology for the recognition of features derived from unstructured human (police officers, traffic safety officers, pedestrians, etc.) motions used in self-driving cars | Yonggu Lee | 2103.2 |
| Daegu | Industry | Spring Cloud Co., Ltd. | Driving situations that the process of assessment and verification of SW platform | Youngki Song | 3972.9 |
| | Academia | Daegu Gyeongbuk Institute of Science and Technology (DGIST) | Core technology development of autonomous cars | Wooyoung Jung | 640.3 |
| | Others | Korea Intelligent Automotive Parts Promotion Agency | Development of validation techniques based on real roads for reliability evaluation of the autonomous vehicle systems | Kyungsoo Yoon | 13,008.5 |

**Table 7.** *Cont.*

| Region | Type of Organization | Organization | R&D Title | Project Manager | Funding (USD Thousand) |
|---|---|---|---|---|---|
| Daejeon | Industry | Dyin Industries Co., Ltd. | Development of ultra-lightweight and low-cost LiDAR sensor module | Deokbae Lee | 2484.7 |
| | Academia | Korea Advanced Institute of Science and Technology (KAIST) | Development of Deep Learning-Based Future Prediction and Risk Assessment technology considering Inter-vehicular Interaction in Cut-in Scenario | Dongseok Keum | 2364.9 |
| | Institutes | Electronics and Telecommunications Research Institute (ETRI) | Development of Driving Computing System Supporting Real-time Sensor Fusion Processing for Self-Driving Car | Seonghoon Kim | 8166.7 |
| | Others | Center for Integrated Smart Sensors | Application of smart camera system | Jongmin Kyung | 11,539.6 |
| Seoul | Industry | Dongwoon Anatech | Development of Next-Generation Smart Camera DSLR-level Multi-Eyes Driving SoC | Jin Park | 5616.5 |
| | Academia | Hanyang University | Development of Traffic Jam Assist System for EV based on Affordable Sensors | Myungho Sunwoo | 2337.0 |
| | Institutes | Korea Institute of Science and Technology (KIST) | Development of complex cognitive core technology for optimal identification and inference under spatio-temporal-view changes | Ik-jae Kim | 4498.3 |
| Sejong | Academia | Hongik University | Deep learning-based array antenna multiple object location tracking using time series correlation and its generalization | Dosik Yoo | 114.6 |
| Ulsan | Academia | Ulsan National Institute of Science and Technology (UNIST) | Research on a flash-type high-accuracy high-resolution CMOS LiDAR sensor with background light suppression | Seongjin Kim | 250.8 |
| Incheon | Industry | Carnavicom Co., Ltd. | The Development of the 8-channel 15 f/s grade scanning LiDAR for autonomous car | Jongtaek Jeong | 1475.5 |
| Jeollabuk-do | Academia | Chonbuk National University | Development of Artificial Intelligence for Autonomous Farming via Deep Learning of Semantic Graphics | Hyungseok Kim | 166.7 |

**Table 7.** *Cont.*

| Region | Type of Organization | Organization | R&D Title | Project Manager | Funding (USD Thousand) |
|---|---|---|---|---|---|
| Chungcheongnam-do | Industry | Dongoh Precision Co., Ltd. | Development of technology to recognize characteristics and location of surrounding sounds for safety driving of autonomous vehicles | Dongran Shin | 96.9 |
| | Academia | Hoseo University | A Fundamental Study of the Optical Camera Communication-based Autonomous Vehicles | Byungwook Kim | 41.7 |
| | Institutes | Korea Automotive Technology Institute (KATECH) | The development of future technology for mixed traffic intersection integrated environmental information regarding the autonomous at the crossroads of the city | Jungwook Lee | 1111.7 |
| Chungcheongbuk-do | Industry | Power Logics Co., Ltd. | Development of automotive camera modules of 1.3MP or higher for mirrorless cars and dash cams and development of smart manufacturing systems with linear position accuracy of $\pm 1$ μm | Jeonwhan Choi | 2333.3 |

## 4. Discussion and Conclusions

### 4.1. Discussion for Collaborative Trans-Regional R&D Strategy on Future Mobility

The proposed framework for a collaborative trans-regional R&D strategy provides a variety of information to implement the two goals of reducing emissions (Korean New Deal) and regional inequality (Region New Deal) in terms of regional, technological, and organizational dimensions. It is developed by considering the important functions of the systematic framework—an evidence-based situation analysis on particular sectors and technologies, a base for investment monitoring, and process management for inclusive communication among stakeholders [50]. In order to demonstrate the utilization of the framework, we established four research questions (RQ) (two sub-research questions in RQ 2 and RQ3). First, based on RQ1, we revealed the overall status of regional government investment in future mobility from the perspective of automotive company locations during 2015–2020. This information may be a good starting point for a range of stakeholders to discuss the appropriateness of national R&D investment for the improvement of regions in the context of reliable sources and objective analysis. Second, based on RQ2-1 and RQ2-2, we presented the distribution and trends of investment in future mobility-related technology areas during 2015–2020. Third, based on RQ3-1 and RQ3-2, we showed various pieces of information to understand the comprehensive competitiveness of future mobility-related fields from the viewpoint of regions. For the central and local stakeholders, the investment information based on both the seven classified future mobility-related fields and the 17 regions in this study can be adopted as a medium that facilitates communication among them to discuss leading local R&D organizations for regional specialization. Finally, based on RQ4-1 and RQ4-2, we indicated the key investing local organizations and their R&D activities in future mobility-related technology areas, including academia, industry, and research institutes. This information may serve the potential trans-regional collaborative R&D partners or eligible experts and members of the strategy committee due to its objective, transparent nature, and legitimacy of the selection process.

The framework empirically applied in this paper provides useful information and implications for establishing R&D strategies for various stakeholders. First, for stakeholders related to central and local governments, it provides basic information on the directions and strategy of various R&D support policies to secure balanced national development and regional industrial competitiveness and sustainable competitive advantage. In order for the government to establish detailed R&D policies and implementation strategies, it is necessary to identify the R&D portfolio that shows the positioning of the detailed R&D areas that each region is leading in specific industries and the status of innovative organizations and companies with technological development competitiveness in the region. In order to secure technological competitiveness and human and material resources, local governments should establish a wide cooperation strategy with innovative organizations with a competitive advantage. Therefore, the framework can be used as a policy tool to foster and support the R&D of central and local governments.

The framework would also be useful for innovation organizations to expand their roles in their unique R&D and business areas and pursue their own benefits. Universities and research institutes can utilize the framework in various fields, such as ensuring continuous research opportunities, expanding science and technology infrastructure, training suitable for technology commercialization, and providing employment opportunities. In the case of industries, business benefits such as commercialization through technology transfer from universities and research institutes, exchange and securing of high-quality human resources, and expansion of applied technologies can be attained. Further, it can be used as a useful tool to create synergy effects through mutual R&D collaboration and the pursuit of the unique interests of innovative organizations.

As discussed in Section 1.1.5, the impact of the COVID-19 pandemic has encouraged people to return to their cars for their commute rather than using public transportation (i.e., bus and train) because of safety concerns [48,51,52]. At the same time, more people changed their perceptions and preferred using sustainable mobility modes to protect the

climate [52]. In order to meet the current and future needs, the strategy committee must consider an increase in national R&D funding in some technical fields, including Cluster 1 (core electric/electronic sub-assembly for green cars) and Cluster 3 (hydrogen charging infrastructure), to accelerate the electrification of public transport faster than the targeted year. Owing to increased energy consumption and congestion in the urban environment, increased government investment in public transport services and the adoption of a connected and integrated system across mobility modes, including active and shared mobility solutions (Cluster 4, autonomous vehicle service platform), are strongly recommended. In addition, it is required for the committee to regain the public's confidence in public transport through more investment, such as safer and cleaner trains and buses. The new R&D agenda for this new societal requirement needs to be discussed.

*4.2. Conclusions*

The Korean government recently proposed a new industrial strategy called the Korean New Deal, which combines climate change mitigation (Green New Deal) and the elimination of economic inequality (Regionally Balanced New Deal) to ensure sustainable development in response to the COVID-19 pandemic. Many scholars have asserted that a strategy for the green economic transition requires a long-term commitment to public support and funding for green R&D and innovation [27]; this agrees with the need for a fine-tuned analytical framework that considers individual regional variations caused by differences in the endowment of relevant assets such as natural resources, technologies, qualifications and skills, and institutional factors [8,27,33]. However, there are no practical examples of such a framework for industrial strategies that support the exchange of knowledge and other assets within and beyond regions by establishing horizontal and vertical coordination among the various stakeholders, actor networks, and policy agents at different spatial scales (from local to central government) [33]. To the best of our knowledge, this study represents the first empirical attempt to provide a framework for a precise trans-regional innovation scheme with regional, technological, and organizational dimensions.

Therefore, the study has two important contributions. First, we present a procedure and framework for analyzing the detailed status and trends of integrated national industry strategies that combine climate change mitigation (Green New Deal) and the elimination of economic inequality (Regionally Balanced New Deal). Specifically, this study adopted nationally funded research project data that contain the aim and contents of projects, the project periods, and the amount of funding rather than paper and/or patent data. This allows us to objectively classify the research fields of projects according to a specific industry (i.e., future mobility) based on the ASJC codes, and present highly relevant information that can support policy makers and R&D program designers in making more well-informed decisions regarding the current scientific capabilities of individual research fields. These data, including the scale of investment, are normally very difficult to deduce when conducting a scientific publication or patent-based analysis. Moreover, this information can be used as a framework for stakeholders (from a wide variety of governmental and academic institutions and industries) to coordinate the development of green technology-related projects from the dual perspective of green technology clusters and regions.

The second major contribution is that this study demonstrates how to operate the framework based on nationally funded research projects on the national level in relation to the Eco-Friendly Mobility of the Future project, which is one of the most important projects in the Green New Deal. Moreover, we illustrate the level of investment in nationally funded research projects related to future mobility for each region in Korea and different technology clusters of future mobility, as well as the role of different research organizations in each technology cluster and each region during 2015−2020. Although the CR has considerably higher technological abilities than other regions, the results of this study explicitly verify the comparable innovation capabilities of all 17 regions for seven distinct technology areas and indicated the differences in individual technology areas for each region. Our results indicate that the CR and Daejeon areas have high innovation capability

in most research areas, whereas some regions have relatively stronger capabilities in specific technology areas such as hydrogen infrastructure. These results represent empirical evidence for the differentiation of regional competitiveness. Moreover, we systematically list the organizations in future mobility-related technology areas that may serve as trans-regional collaborative R&D partners from a regional perspective.

### 4.3. Limitations and Further Research

Despite these contributions, this study also has some limitations that present challenging questions for future research. One concern is that only nationally funded research project data from the central government were utilized. Although R&D expenditure determined by individual local governments is less than that of the central government, local government-led research project data also exist. Therefore, it is necessary to encompass all national funding data from both central and local governments in order to accurately understand the current status of the 17 regions of Korea. However, the information provided in this study in terms of technology clusters can become primary data, allowing local governments to integrate their own R&D expenditure for future mobility. Another limitation that can be corrected by future studies is the lack of funding data from the US, EU, and other nations, which could be used to conduct a comparative analysis of the absolute amount of R&D funding for each technology cluster among different nations. Future studies employing such a methodology can support national industrial strategies of future mobility-related R&D project portfolio management.

**Author Contributions:** Conceptualization, D.L.; data curation, D.L.; formal analysis, D.L.; funding acquisition, K.K.; investigation, D.L.; methodology, K.K.; project administration, K.K.; resources, D.L.; software, K.K.; supervision, K.K.; validation, K.K.; visualization, D.L. and K.K.; writing—original draft, D.L. and K.K.; writing—review and editing, K.K. Both authors have read and agreed to the published version of the manuscript.

**Funding:** This research was supported by the Korea Institute of Science and Technology Information (KISTI) of the Ministry of Science and ICT, Korea (MSIT) (No. K-20-L03-C03-S01: Artificial intelligence-based decision making for supporting national R&D policy and No. K-21-L03-C04-S01: Advancement of innovation strategy analysis models for science, technology, and industry).

**Institutional Review Board Statement:** Not applicable.

**Informed Consent Statement:** Not applicable.

**Data Availability Statement:** Not applicable.

**Conflicts of Interest:** The authors declare no conflict of interest.

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
