# Peer review of "A Collaborative Trans-Regional R&D Strategy for the South Korea Green New Deal to Achieve Future Mobility"

_sustainability, doi:10.3390/su13158637_

Round 1
Reviewer 1 Report
Dear authors
The article is designed really well. Methodology, contribution, and research gap are obvious. I feel your article can be easily accepted. The only tip that can make your article better is the readership of the title and the keywords. The title and keywords are not supporting each other. It's not common to have many differences between these two.
Author Response
Point-by-point responses to reviewers’ comments (detail the revisions)
We wish to thank the reviewers for reviewing our manuscript, Sustainability-1305810 titled "A Collaborative Trans-regional R&D Strategy for the South Korea Green New Deal to Achieve Future Mobility," and providing the comments and suggestions to improve the quality of the manuscript.
We are grateful for the opportunity to submit a revised version of our manuscript and sincerely thank the reviewers for their constructive criticism and helpful comments that we have used to improve the manuscript.
Below we have responded to all comments made by the reviewers.
Author's Response to Reviewer 1’s comments
Comments and Suggestions for Authors:
The article is designed really well. Methodology, contribution, and research gap are obvious. I feel your article can be easily accepted. The only tip that can make your article better is the readership of the title and the keywords. The title and keywords are not supporting each other. It's not common to have many differences between these two.
Point 1: The only tip that can make your article better is the readership of the title and the keywords.
Response 1: The reviewer raises an important point. We agree with the reviewer’s comment. The keywords have been revised accordingly (please see below) [Page 1, line 22-23].
(Before)
Keywords: Korean Green New Deal; Regional inequality; Electric vehicles; Autonomous vehicles; Nationally funded project data |
(After)
Keywords: Korean Green New Deal; Collaboration; Trans-regional R&D strategy; Future mobility; Nationally funded project data; Framework |
Reviewer 2 Report
- It is suggested to achieve superior clarity of the paper purpose by higher coherence between the paper title (“… collaborative trans-regional R&D strategy …”) and Abstract (“… framework for trans-regional innovation approach …”) – e.g.:
- "collaborative strategy" versus "framework";
- "R&D" versus "innovation".
- More coherence between the above (on one side) and the keywords (on the other) is also recommended.
- Ultimately, which is that “Collaborative Trans-regional R&D” Strategy?
- As far as the “dual aims” [row 12]: the authors should elaborate and explain how their convergence is realised, upfront.
- The research results are mostly about the new future mobility-related technologies used in autonomous vehicles. It is strongly recommended to document and explain how the concept of “autonomous vehicle” is related to the concept of “sustainability” (the journal’s profile).
- It is strongly recommended to present the research results in strict correspondence to the research questions. If the case, the list of research questions should be revisited.
- As far as research objectives: (RQ3) should be split – as it includes several (related but different) questions.
- The major trends identified as result of research (under RQ2) should be highlighted and the corresponding recommendations systematically presented.
- The practical implications of the research should be clearly presented – specifically by categories of stakeholders.
- The authors have to highlight their original contribution and advance – as compared to the state-of-the-art literature.
- The sub-section 3.2.4 presents lots of data and numbers. What are they useful for – What’s the idea to be supported by those numbers? More systematic and synthetic presentation is recommended.
- Some graphic materials (e.g. Figure 2, Figure 3, Figure 9) should be redesigned – in order to be reasonably more legible.
- Figure 1 [rows 124-125]: The abbreviations (MSIT, MOIS, MOE, etc.) should be completely defined.
- Figures 2 and 9 should be ‘Tables’, actually.
- While reviewing the proposed paper, some phrases should be more explicit: e.g. “Various developed unions …” [row 40] – What kind of “unions”? Because the term “union” has lots of meanings.
- It is also suggested to revisit the English language (e.g. “… capital region and Daejon, has …” [row 19] et al.) as well as spelling errors (e.g. “… 2015-2010 …” [row 164] et al.)
Author Response
Author's Response to Reviewer 2’s comments
Comments and Suggestions for Authors
Point 1: It is suggested to achieve superior clarity of the paper purpose by higher coherence between the paper title (“… collaborative trans-regional R&D strategy …”) and Abstract (“… framework for trans-regional innovation approach …”) – e.g.:
- "collaborative strategy" versus "framework";
- "R&D" versus "innovation".
Response 1: The reviewer raises an important point. We agree with the reviewer’s comment. By matching the keywords of the abstract with the keywords of the title of the revised manuscript, according to the reviewer’s comments, the consistency between the title and the abstract was improved (please see below) [Page 1, line 12-14].
(Before)
In order to accomplish these dual aims, this study provides a framework for a precise trans-regional innovation approach with three key dimensions: regional, technological, and organizational. |
(After)
To accomplish these dual aims, this study provides a collaborative trans-regional R&D strategy and a precise framework with three key dimensions: regional, technological, and organizational. |
Point 2: More coherence between the above (on one side) and the keywords (on the other) is also recommended.
Response 2: As mentioned in “Response 1”, by matching the keyword of title with the keyword of abstract and main keywords, we tried to increase the overall consistency. The keywords have been revised according to the reviewer’s comment (please see below) [Page 1, line 22-23].
(Before)
Keywords: Korean Green New Deal; Regional inequality; Electric vehicles; Autonomous vehicles; Nationally funded project data |
(After)
Keywords: Korean Green New Deal; Collaboration; Trans-regional R&D strategy; Future mobility; Nationally funded project data; Framework |
Point 3: Ultimately, which is that “Collaborative Trans-regional R&D” Strategy?
Response 3: Reviewers raise important points. We agree with the reviewers and sincerely appreciate this recommendation. According to the reviewer's opinion, we had an in-depth discussion on “Collaborative Trans-regional R&D strategy,” and this important discussion is faithfully reflected in an independent section of the “discussion and conclusion” in the revised manuscript (please see below) [Page 25-26, line 651-715].
[Page 25-26, line 651-715]. 4.1. Discussion for collaborative trans-regional R&D strategy on future mobility The proposed framework for a collaborative trans-regional R&D strategy provides a variety of information to implement the two goals of reducing emission (Korean New Deal) and regional inequality (Region New Deal) in terms of regional, technological, and organizational dimensions. It is developed by considering the important functions of the systematic framework — an evidence-based situation analysis on particular sectors and technologies, a base for investment monitoring, and process management for inclusive communication among stakeholders [51]. In order to demonstrate the utilization of the framework, we established four research questions (RQ) (2 sub-research questions in RQ 2 and RQ3). First, based on RQ1, we revealed the overall status of regional government investment in future mobility from the perspective of automotive company locations during 2015–2020. This information may be a good starting point for a range of stakeholders to discuss the appropriateness of national R&D investment for the improvement of regions in the context of reliable sources and objective analysis. Second, based on RQ2-1 and RQ2-2, we presented the distribution and trends of investment in future mobility-related technology areas during 2015–2020. Third, based on RQ3-1 and RQ3-2, we showed various pieces of information to understand the comprehensive competitiveness of future mobility-related fields from the viewpoint of regions. For the central and local stakeholders, the investment information based on both the seven classified future mobility-related fields and the 17 regions in this study can be adopted as a medium that facilitates communication among them to discuss leading local R&D organizations for regional specialization. Finally, based on RQ4-1 and RQ4-2, we indicated the key investing local organizations and their R&D activities in future mobility-related technology areas, including academia, industry, and research institutes. This information may serve the potential trans-regional collaborative R&D partners or eligible experts and members of the strategy committee due to its objective, transparent nature, and legitimacy of the selection process.
The framework empirically applied in this paper provides useful information and implications for establishing R&D strategies for various stakeholders. First, for stakeholders related to central and local governments, it provides basic information on the directions and strategy of various R&D support policies to secure balanced national development and regional industrial competitiveness and sustainable competitive advantage. In order for the government to establish detailed R&D policies and implementation strategies, it is necessary to identify the R&D portfolio that shows the positioning of the detailed R&D areas that each region is leading in specific industries and the status of innovative organizations and companies with technological development competitiveness in the region. In order to secure technological competitiveness and human-material resources, local governments should establish a super-wide cooperation strategy with innovative organizations with a competitive advantage. Therefore, the framework can be used as a policy tool to foster and support the R&D of central and local governments.
The framework would also be useful for innovation organizations to expand their roles in their unique R&D and business areas and pursue their own benefits. Universities and research institutes can utilize the framework in various fields, such as ensuring continuous research opportunities, expanding science and technology infrastructure, training suitable for technology commercialization, and providing employment opportunities. In the case of industries, business benefits such as commercialization through technology transfer from universities and research institutes, exchange and securing of high-quality human resources, and expansion of applied technologies can be attained. Further, it can be used as a useful tool to create synergy effects through mutual R&D collaboration and the pursuit of the unique interests of innovative organizations.
As discussed in subsection 1.1.5, the impact of the COVID-19 pandemic has encouraged people to get back into their cars for their commute rather than using public transportation (i.e., bus and train) due to safety concerns [49, 52, 54]. At the same time, more people increased their perceptions of using sustainable mobility modes to protect the climate [54]. In order to meet the current and future needs, the strategy committee must consider the increase of national R&D funding in some technical fields, including Cluster 1. Core electric/electronic sub-assembly for green cars and Cluster 3. Hydrogen charging infrastructure, to accelerate the electrification of public transport faster than the targeted year. Owing to increased energy consumption and congestion in the urban environment, increased government investment in public transport services and adoption of a connected and integrated system across transport modes, including active and shared mobility solutions (Cluster 4. Autonomous vehicle service platform) is strongly recommended. In addition, it is required for the committee to regain the public’s confidence in public transport by more investment in public transport such as safer and cleaner trains and buses. The new R&D agenda for this new societal requirement needs to be discussed.
|
Point 4: As far as the “dual aims” [row 12]: the authors should elaborate and explain how their convergence is realized, upfront.
Response 4: We sincerely appreciate the reviewer for suggesting this. According to the reviewer’s comments, we elaborated and explained the “dual aims” in the “introduction” section of the revised manuscript (please see below) [Page 2, line 71-88, Page 3, line 116-124, Page 5-6, line 234-243].
[Page 2, line 71-88] Several studies have devoted to the normative statement that the Korean government endeavors to accomplish both the goal of reducing greenhouse gas emissions via the Green New Deal and that of reducing deepening regional inequality via the Regionally Balanced New Deal through effective collaborations with other stakeholders such as private companies, academia, research institutes, and agencies owing to inter-linked concerns over climate change, air pollution, energy security and the global competitiveness in the key industries [8, 10, 21]. However, studies on developing a systematic framework for collaborative trans-regional R&D strategy planning are lacking. In an attempt to bridge this gap, we proposed a systematic framework that would support the organization committee that consists of stakeholders who implement the national green industrial strategy by providing valuable and detailed information on relevant evidence-based situations and inclusive, relevant local implementation actors who may become the collaborative partners and/or members of the committee in a particular technology sector. Moreover, we applied the proposed framework on future mobility, one of the five key projects of the Green New Deal, to induce the implications for a collaborative trans-regional R&D strategy plan. Meanwhile, we investigated the global changes in public transport stemmed from the COVID-19 pandemic to discuss the directions to the development of future mobility.
[Page 3, line 116-124] Thus, it is important to closely combine the aim of reducing greenhouse gas emissions (from the Korean Green New Deal) with that of reducing deepening regional inequality (from the Regionally Balanced New Deal) by establishing a national collaborative R&D strategy for a trans-regional innovation approach that will accelerate the creation, dissemination, absorption, and application of new scientific and technological knowledge and ensure inter-organizational linkages across regions [21, 33]. Such a strategic approach allows regional actors to promote collaborative scientific and technological projects across Korea, which can gather distant partners and provide opportunities to further develop capabilities in their areas of specialization [25]. .
[Page 5-6, line 234-243]. These previous studies [8, 10, 21, 27, 31, 33, 41, 44] asserted only the normative statements that planning a collaborative strategy with a trans-regional perspective is important to successfully accomplish these green innovation initiatives and that the existing strategy lacks an explicit framework. Therefore, it is necessary to build a systematic (investment) framework for a fine-tuned trans-regional innovation scheme with regional, technological, and organizational dimensions, thereby identifying the reasons for the gap in a regional variation of innovation capabilities and then suggesting appropriate strategies to bridge the gap. The proposed framework is established on the abovementioned consensus on directions for a better, coherent/R&D strategy (planning) to facilitate collaborations with a range of stakeholders via continuous communication.
|
Point 5: The research results are mostly about the new future mobility-related technologies used in autonomous vehicles. It is strongly recommended to document and explain how the concept of “autonomous vehicle” is related to the concept of “sustainability” (the journal’s profile).
Response 5: We sincerely thank the reviewer for suggesting this. Following the reviewer’s comment, how the concept of "autonomous vehicle" relates to the concept of "sustainability" (journal profile) was explained in a separate paragraph in the “introduction” section of the revised manuscript (please see below) [Page 4, line 175-186].
Two zero-emission vehicle technologies—battery electric vehicles and fuel cell (electric) vehicles—have emerged as the main pillars of future mobility [41, 42]. Although the battery electric vehicle market is currently dominating in several countries such as the US, EU, China, and Korea, industrial and government efforts to spur the market development of fuel cell transport are supported by numerous advantages relative to batteries, including refueling times roughly comparable to gasoline, longer driving ranges, fewer space requirements for hydrogen refueling stations, less performance deterioration from battery aging, and less reliance on lithium and cobalt supply chains [39, 42]. Meanwhile, autonomous technologies can improve road safety, increase fuel efficiency and reduce emissions, and improve urban public transportation via multimodal transportation services [43]. Therefore, electric vehicles are likely to integrate autonomous technologies and would play an important role in promoting zero-emission and sustainable transportation. |
Point 6: It is strongly recommended to present the research results in strict correspondence to the research questions. If the case, the list of research questions should be revisited.
Response 6: We thank for the reviewer for suggesting this. According to the reviewer’s comments, we described the research results according to the research questions in the “results” section, and also described the implications by reminding the research questions again in the “discussion” section (please see below) [Page 15-16, 18, 21, 23, 25].
[Page 15, line 464-494] 3.2. Status of government investment in future mobility 3.2.1. Status of government-funded projects according to regions The relationship between the investment status and the automobile industry in each region was then investigated. The information about the investment proportion of future mobility-related research in the 17 regions of Korea may enable a range of stakeholders to provide opportunities for discussions regarding an appropriate R&D investment for the improvement of these regions. [Page 16, line 498-515] 3.2.2. Status and trend of government-funded projects according to technology clusters Figure 6 shows the total amount of national research funding for future mobility in terms of technology clusters. A large proportion of the total national research funding was given to Cluster 2 (intelligent vehicle communication, US$ 334 million, 26%), followed by Cluster 1 (core electric/electronic sub-assembly for green cars, US$ 245 million, 19%). Small amounts of funding were invested in Cluster 4 (autonomous vehicle service platform, US$ 88 million, 7%) and Cluster 3 (hydrogen charging infrastructure, US$ 100 million, 8%). Because Cluster 1 and 2 are considered the core technology areas for improving the performance of future mobility, considerable investment in these areas is expected [31, 62]. It is important for stakeholders to determine the optimal portfolio of R&D projects to maximize the long-term development strategy by comparing heterogeneous R&D projects that are interrelated and aligned with strategies [63]. Hence, it is prerequired to classify projects for facilitating the process of prioritizing R&D projects [64]. This study presents the systematic analysis procedure to classify future mobility-related national projects into technology groups and then provides information about the investment trends of individual technology groups (clusters) to the stakeholders. Thus, the framework opens the starting point to discuss the future of the R&D portfolio to facilitate communication and ensure understanding among them.
[Page 18, line 541-545] 3.2.3. Status of government-funded projects according to technology clusters and regions We also investigated the status of nationally funded projects according to technology clusters and regions to identify the strength of regional technological competitiveness, as shown in Table 5.
[Page 21, line 572-593] 3.2.4. Status of government-funded projects according to technology clusters, regions, and organization types Next, we investigated the status of investment according to technology clusters, regions, and organization type to establish the potential collaborative network between academia, industry, and research institutes in the national future mobility industry (Table 6). Individual regional variations are caused by differences depending on their growth path [28]. Thus, it is hard for regions with limited skills and assets in technology and organizations to overcome such limitations [13]. In order to effectively reduce the difficult conditions, it is needed to understand which types of organizations play a leading role in creating knowledge in specific research fields. It may become the regional base of collaboration with private companies in regions with less-favored research and innovation systems. This study allows stakeholders to consider regional research strategies that re-create networks of industry, universities, and research institutes in low-innovative regions. Table 6 is a regional R&D portfolio showing how much investment in R&D areas based on technology represented by each of the seven clusters has been made from the perspective of 17 regions in Korea, indicating where each area is leading with high competitiveness. In addition to the technical area and the regional perspectives, it is subdivided by an R&D innovation organization. These results show the types of competitive organizations leading technological development related to specific clusters in each region and the characteristics of regional R&D. As for the specific funding size and investment ranking, the industry was the greatest source of investment in Gyeonggi-do ($ 245.7 million), followed by Seoul ($ 124.8 million) and Incheon ($ 47.2 million)
[Page 23, line 624-641] 3.2.5. Potential national collaborative research organizations according to technology clusters Many nations have launched multiple initiatives and endeavored to build R&D collaboration strategies for dealing with regional inequality. Such endeavors require policymakers or national R&D program directors to have detailed information in order to create a task force for policy issues or promote trans-regional collaborative R&D.
The central government of Korea significantly increased some incentives for private R&D organizations in low-innovative regions to participate in national R&D projects by establishing knowledge links through industry-academia-institutes collaborations [28]. For companies in low-innovative regions that look for a potential partner, the government must provide detailed information about what organizations did what R&D projects in a specific technology field. In addition, the pre-condition for developing a coherent/R&D strategy (planning) is to establish a process for inclusive communication among stakeholders [51]. For private R&D organizations in low-innovative regions, this study provides detailed information about the potential partners in a specific research field. For the central-local government, this study presents information about eligible experts in academia, research institutes, and industry who may participate in an ongoing communication process.
[Page 25, line 651-677] 4.1. Discussion for collaborative trans-regional R&D strategy on the future mobility The proposed framework for a collaborative trans-regional R&D strategy provides a variety of information to implement the two goals of reducing emission (Korean New Deal) and regional inequality (Region New Deal) in terms of regional, technological, and organizational dimensions. It is developed by considering the important functions of the systematic framework — an evidence-based situation analysis on particular sectors and technologies, a base for investment monitoring, and process management for inclusive communication among stakeholders [51]. In order to demonstrate the utilization of the framework, we established four research questions (RQ) (2 sub-research questions in RQ 2 and RQ3). First, based on RQ1, we revealed the overall status of regional government investment in future mobility from the perspective of automotive company locations during 2015–2020. This information may be a good starting point for a range of stakeholders to discuss the appropriateness of national R&D investment for the improvement of regions in the context of reliable sources and objective analysis. Second, based on RQ2-1 and RQ2-2, we presented the distribution and trends of investment in future mobility-related technology areas during 2015–2020. Third, based on RQ3-1 and RQ3-2, we showed various pieces of information to understand the comprehensive competitiveness of future mobility-related fields from the viewpoint of regions. For the central and local stakeholders, the investment information based on both the seven classified future mobility-related fields and the 17 regions in this study can be adopted as a medium that facilitates communication among them to discuss leading local R&D organizations for regional specialization. Finally, based on RQ4-1 and RQ4-2, we indicated the key investing local organizations and their R&D activities in future mobility-related technology areas, including academia, industry, and research institutes. This information may serve the potential trans-regional collaborative R&D partners or eligible experts and members of the strategy committee due to its objective, transparent nature, and legitimacy of the selection process.
|
Point 7: As far as research objectives: (RQ3) should be split – as it includes several (related but different) questions.
Response 7: We thank the reviewer for suggesting this. According to the reviewer’s comments, we described six research questions in the revised manuscript compared to the three research questions in the previous manuscript version. To explain in detail, RQ2 was separated into two RQs, and RQ3 was separated into three RQs (two sub-RQs and one independent RQ4) (please see below) [Page 6-7, line 282-306].
(Before)
(RQ1) How much did the Korean government invest in regional future mobility from the perspective of automotive company locations during the period 2015–2020? (RQ2) What was the distribution of investment in future mobility-related technology areas in 2015–2020, and what trends of investment in technology areas emerged during this period? (RQ3) What was the regional distribution of investment in future mobility-related technology areas? What types of organization (among academia, industry, and research institutes) have played an important role in future mobility-related technology areas from a regional perspective? What organizations related to future mobility-related technology areas may serve as trans-regional collaborative R&D partners from a regional perspective? |
(After)
Our primary research question was, Research Question RQ1: How much did the Korean government invest in regional future mobility from the perspective of automotive company locations during the period 2015–2020?
In addition to understanding the status and trends of investment of the Korean government in future mobility-related technology areas, we examined the following research questions.
Research Question RQ2-1: What was the distribution of investment in future mobility-related technology areas in 2015–2020?
Research Question RQ2-2: What trends of investment in technology areas emerged during this period?
To provide the information for comprehending the competitiveness of future mobility-related fields in terms of regions, we investigated the status and trends of investment of Korean government by asking the following research questions:
Research Question RQ3-1: What was the regional distribution of investment in future mobility-related technology areas?
Research Question RQ3-2: What types of organizations (among academia, industry, and research institutes) have played an important role in future mobility-related technology areas from a regional perspective? Finally, we looked closely at the detailed research activities in future mobility-related research fields for information about potential partners who may share the knowledge among other stakeholders and asked the following research questions. Research Question RQ4: What organizations related to future mobility-related technology areas may serve as trans-regional collaborative R&D partners from a regional perspective? |
Point 8: The major trends identified as result of research (under RQ2) should be highlighted and the corresponding recommendations systematically presented.
Response 8: According to the reviewer’s comments, we described the research results according to the research questions in the “results” section (3.2.2. Status and trend of government-funded projects according to technology clusters) [Page 16-17, line 498-539].
Point 9: The practical implications of the research should be clearly presented – specifically by categories of stakeholders.
Response 9: We agree with the reviewer and sincerely appreciate this recommendation. According to the reviewer's comments, we clearly presented the practical meaning and implications of this study by stakeholder category in a separate paragraph of “discussion and conclusion” of the revised manuscript (please see below) [Page 25-26, line 678-700].
The framework empirically applied in this paper provides useful information and implications for establishing R&D strategies for various stakeholders. First, for stakeholders related to central and local governments, it provides basic information on the directions and strategy of various R&D support policies to secure balanced national development and regional industrial competitiveness and sustainable competitive advantage. In order for the government to establish detailed R&D policies and implementation strategies, it is necessary to identify the R&D portfolio that shows the positioning of the detailed R&D areas that each region is leading in specific industries and the status of innovative organizations and companies with technological development competitiveness in the region. In order to secure technological competitiveness and human-material resources, local governments should establish a super-wide cooperation strategy with innovative organizations with a competitive advantage. Therefore, the framework can be used as a policy tool to foster and support the R&D of central and local governments. The framework would also be useful for innovation organizations to expand their roles in their unique R&D and business areas and pursue their own benefits. Universities and research institutes can utilize the framework in various fields, such as ensuring continuous research opportunities, expanding science and technology infrastructure, training suitable for technology commercialization, and providing employment opportunities. In the case of industries, business benefits such as commercialization through technology transfer from universities and research institutes, exchange and securing of high-quality human resources, and expansion of applied technologies can be attained. Further, it can be used as a useful tool to create synergy effects through mutual R&D collaboration and the pursuit of the unique interests of innovative organizations.
|
Point 10: The authors have to highlight their original contribution and advance – as compared to the state-of-the-art literature.
Response 10: We sincerely appreciate the reviewer for suggesting this. This comments significantly improved the quality and completeness of our manuscript. In response to the reviewer’s comments, we described our original contribution and advance in a separate section of “Introduction (1.1. Theoretical background and literature review)” of the revised manuscript. “1.1. Theoretical background and literature review” consists of a total of five subsections. After reviewing each previous literature in five sections, we emphasized the original contribution and significance of this study by comparing it with previous literature [Page 2-6, line 90-266].
Point 11: The sub-section 3.2.4 presents lots of data and numbers. What are they useful for – What’s the idea to be supported by those numbers? More systematic and synthetic presentation is recommended.
Response 11: We thank the reviewer for suggesting this. According to the reviewer's comment, the results of subsection 3.2.4 were described more systematically and comprehensively (please see below). [Page 21, line 572-593].
3.2.4. Status of government-funded projects according to technology clusters, regions, and organization types Next, we investigated the status of investment according to technology clusters, regions, and organization type to establish the potential collaborative network between academia, industry, and research institutes in the national future mobility industry (Table 6). Individual regional variations are caused by differences depending on their growth path [28]. Thus, it is hard for regions with limited skills and assets in technology and organizations to overcome such limitations [13]. In order to effectively reduce the difficult conditions, it is needed to understand which types of organizations play a leading role in creating knowledge in specific research fields. It may become the regional base of collaboration with private companies in regions with less-favored research and innovation systems. This study allows stakeholders to consider regional research strategies that re-create networks of industry, universities, and research institutes in low-innovative regions.
Table 6 is a regional R&D portfolio showing how much investment in R&D areas based on technology represented by each of the seven clusters has been made from the perspective of 17 regions in Korea, indicating where each area is leading with high competitiveness. In addition to the technical area and the regional perspectives, it is subdivided by an R&D innovation organization. These results show the types of competitive organizations leading technological development related to specific clusters in each region and the characteristics of regional R&D. As for the specific funding size and investment ranking, the industry was the greatest source of investment in Gyeonggi-do ($ 245.7 million), followed by Seoul ($ 124.8 million) and Incheon ($ 47.2 million)… |
Point 12: Some graphic materials (e.g. Figure 2, Figure 3, Figure 9) should be redesigned – in order to be reasonably more legible.
Response 12: We thank the reviewer for suggesting this. According to the reviewer’s comment, “Figure 2” and “Figure 9” have been redesigned to be more legible by changing them to table format, “Figure 3” has been corrected by increasing the resolution [Page 9, 11, 22].
1. Page 9: Figure 2 à Table 1 (changed to table format) 2. Page 11: Figure 3 à Figure 2 (changed to a high resolution file) 3. Page 22: Figure 9 à Table 6 (changed to table format) |
Point 13: Figure 1 [rows 124-125]: The abbreviations (MSIT, MOIS, MOE, etc.) should be completely defined.
Response 13: Thanks for this recommendation. According to the reviewer’s comment, the definition of the abbreviations (MSIT, MOIS, MOE, etc.) was described in the text, and the full name was written instead of the abbreviations (please see below) [Page 4].
Point 14: Figures 2 and 9 should be ‘Tables’, actually.
Response 14: We thank the reviewer for suggesting this. As mentioned in our response to “Point 12”, “Figure 2,” and “Figure 9” have been changed to table format according to the reviewer’s comment (please see below) [Page 9, 22].
1. Page 9: Figure 2 à Table 1 (changed to table format) 2. Page 22: Figure 9 à Table 6 (changed to table format) |
(Before) Figure 2.
|
(After) Table 1.
|
(Before) Figure 9.
|
(After) Table 6.
|
Point 15: While reviewing the proposed paper, some phrases should be more explicit: e.g. “Various developed unions …” [row 40] – What kind of “unions”? Because the term “union” has lots of meanings.
Response 15: Thanks for pointing out the confusion here. We agree with the reviewer’s comment and have corrected the confusing term “Various developed unions” to the more explicit term “Various developed nations” [Page 1, line 40].
Point 16: It is also suggested to revisit the English language (e.g. “… capital region and Daejon, has …” [row 19] et al.) as well as spelling errors (e.g. “… 2015-2010 …” [row 164] et al.)
Response 16: Thanks for pointing out these errors. The spelling errors have been corrected and revised as required (e.g. 2015-2010 à 2015-2020). Meanwhile, “Daejeon” is the name of the city's fifth-largest metropolis located in central South Korea. Therefore, the first letter of Daejeon is capitalized throughout the manuscript.

Reviewer 3 Report
The manuscript addresses a current issue. The methodology adopted is robust and well explored. The context is also introduced. However, the discussion of the results deserves more in-depth work.
Improvement suggestions:
- The introduction section is too long. I suggest the division into two separate sections: Introduction and Literature Review / Background
- The last paragraph of the introduction section should introduce the structure of the manuscript.
- Research questions are too complex, particularly RQ2 and RQ3. Why not divide them into more research questions?
- Figure 2 is not exactly a figure. It is a table. Change it to a table would also improve the quality of resolution.
- It is not clear how the search terms were defined.
- It is not totally clear how the seven clusters are interconnected. Explore better the relationship and dependence among them.
- The impact of COVID-19 in the future of mobility should be better discussed in the discussion section. I suggest the inclusion of the following references:
https://www.sciencedirect.com/science/article/pii/S2590198221000816
https://www.mdpi.com/2071-1050/12/21/8829
https://www.scirp.org/journal/paperinformation.aspx?paperid=107496
- The discussion section is not correctly explored. In fact, the discussion and conclusions section is more related to conclusions than discussion. I recommend the division into two separate sections: discussion; conclusions. The authors need to better describe and explore the discussion section.
Author Response
Author's Response to Reviewer 3’s comments
Comments and Suggestions for Authors:
The manuscript addresses a current issue. The methodology adopted is robust and well explored. The context is also introduced. However, the discussion of the results deserves more in-depth work.
Improvement suggestions:
Point 1: The introduction section is too long. I suggest the division into two separate sections: Introduction and Literature Review / Background
Response 1: The reviewer raises an important point. We appreciate the valuable consideration and feedback from the reviewer. We followed the reviewer's recommendations and tried to reflect them in the revised manuscript faithfully. We believe that this description benefits the understanding and readability of the manuscript [Page 1~7, line 25- 312].
In the revised manuscript, the introduction was divided into two major subsections, and the first subsection was divided into five detailed topics according to other reviewer’s comments. The table of contents of the introduction is as follows.
1. Introduction 1.1. Theoretical background and literature review 1.1.1. Inherent purpose of Korean Green New Deal 1.1.3. Organizational structure of the national strategy 1.1.4. Future mobility policies of Korea 1.1.2. Needs for developing a systematic framework for a collaborative trans-regional R&D strategy plan 1.1.5. Changes in sustainable urban mobility modes in the post-COVID-19 era 1.2. Research purpose and research questions
|
Point 2: The last paragraph of the introduction section should introduce the structure of the manuscript.
Response 2: We thank you for pointing out this. According to the reviewer’s comments, we have revised the introduction section (please see below) [Page 7, line 307-312].
The remainder of this article is structured as follows. Following this general introduction, the “materials and methods” section describes the framework and methodology. The “results” section presents comparative results of the research profiling and machine learning analyses. Finally, the “discussion and conclusion” sections elaborate on the research contributions, implications for practice, and research limitations and indicate promising research opportunities to pursue in the future. |
Point 3: Research questions are too complex, particularly RQ2 and RQ3. Why not divide them into more research questions?
Response 3: We thank the reviewer for suggesting this. According to the reviewer’s comments, we described six research questions in the revised manuscript compared to the three research questions in the previous manuscript version. To be more specific, RQ2 was separated into two RQs, and RQ3 was separated into three RQs (two sub-RQs and one independent RQ4) (please see below) [Page 6-7, line 282-306].
(Before)
(RQ1) How much did the Korean government invest in regional future mobility from the perspective of automotive company locations during the period 2015–2020? (RQ2) What was the distribution of investment in future mobility-related technology areas in 2015–2020, and what trends of investment in technology areas emerged during this period? (RQ3) What was the regional distribution of investment in future mobility-related technology areas? What types of organization (among academia, industry, and research institutes) have played an important role in future mobility-related technology areas from a regional perspective? What organizations related to future mobility-related technology areas may serve as trans-regional collaborative R&D partners from a regional perspective? |
(After)
Our primary research question was, Research Question RQ1: How much did the Korean government invest in regional future mobility from the perspective of automotive company locations during the period 2015–2020? In addition to understanding the status and trends of investment of the Korean government in future mobility-related technology areas, we examined the following research questions. Research Question RQ2-1: What was the distribution of investment in future mobility-related technology areas in 2015–2020? Research Question RQ2-2: What trends of investment in technology areas emerged during this period? To provide the information for comprehending the competitiveness of future mobility-related fields in terms of regions, we investigated the status and trends of investment of Korean government by asking the following research questions:
Research Question RQ3-1: What was the regional distribution of investment in future mobility-related technology areas? Research Question RQ3-2: What types of organizations (among academia, industry, and research institutes) have played an important role in future mobility-related technology areas from a regional perspective? Finally, we looked closely at the detailed research activities in future mobility-related research fields for information about potential partners who may share the knowledge among other stakeholders and asked the following research questions. Research Question RQ4: What organizations related to future mobility-related technology areas may serve as trans-regional collaborative R&D partners from a regional perspective? |
Point 4: Figure 2 is not exactly a figure. It is a table. Change it to a table would also improve the quality of resolution.
Response 4: We thank the reviewer for suggesting this. We changed Figure 2 to table format (Table 1) according to the reviewer’s comment (please see below) [Page 9].
(Before)
|
(After)
|
Point 5: It is not clear how the search terms were defined.
Response 5: Thanks for this recommendation. We have described the definition of the search terms according to the reviewer’s comment (please see below) [Page 7, line 320-327].
[Page 7, line 320-327] Under the guidance of the experts from universities, research institutes, and industries, the two authors conducted the full search strategy and data collection together using the following keywords and the combination of their variants during the search query. Key search terms used included “e-mobility,” “automobility,” “green car,” “electric vehicle,” “electric vehicle with battery,” “electric vehicle fuel cell,” “fuel cell vehicle,” “hydrogen fuel cell vehicle,” “hybrid electric vehicle with fuel cell,” “plug-in hybrid electric vehicle,” “autonomous vehicle,” “self-driving car,” “driverless car,” and “connected autonomous vehicle.” |
Point 6: It is not totally clear how the seven clusters are interconnected. Explore better the relationship and dependence among them.
Response 6: Thanks for this recommendation. This has been revised according to the reviewer’s comment (please see below) [Page 13, line 447-461].
[Page 13, line 447-461 in revised manuscript] The seven clusters reflected the technological directions of the automotive industry that electric vehicles have integrated into the widespread advancement and adoption of alternative fuels, autonomous vehicles, and MaaS [60]. Electric vehicles that draw electricity stored in rechargeable battery packs or a fuel cell powered by hydrogen to power electric motors with motor controllers were classified into Cluster 1. In order to reduce carbon emissions and air pollution; operate at optimal efficiency; increase safety for drivers, passengers, as well as for bicyclists and pedestrians; and enhance the traffic flows and multimodality to decrease urban traffic congestion (Cluster 6 and Cluster 7), many governments play a key role in expanding the electrification of road transport by providing battery charging/hydrogen refueling infrastructure (Cluster 5/Cluster 3) and connectivity (i.e., V2X) infrastructure (Cluster 2) and accelerating the eco-friendly car commercialization with spurring R&D activities such as sustainable mobility services for urban transport with MaaS and/or TaaS (Cluster 4). In the following subsections, we present the status or trend of nationally-funded projects for future mobility in Korea in terms of technology clusters, regions, and organizations. |
Point 7: The impact of COVID-19 in the future of mobility should be better discussed in the discussion section. I suggest the inclusion of the following references:
https://www.sciencedirect.com/science/article/pii/S2590198221000816
https://www.mdpi.com/2071-1050/12/21/8829
https://www.scirp.org/journal/paperinformation.aspx?paperid=107496
Response 7: We thank the reviewer for suggesting this. According to the reviewer's comments and after thoroughly reviewing the above-mentioned references, we faithfully explained and discussed this issue in detail in a separate section of the “introduction” and “discussion” (please see below).
[Page 6, line 245-266 in revised manuscript] 1.1.5. Changes in sustainable urban mobility modes in the post-COVID-19 era The COVID-19 pandemic had swift and brutal impacts on the operation of the current transportation infrastructure [48, 52, 53]. In many nations, safety concerns, anxiety, and stress levels increased in society regarding using public transport after the beginning of the pandemic [52, 54]. In the Republic of Korea, experts expected a mobility modal shift of 94.4% from public transport to personal car and only an expected 5.6% shift to bicycles. Almost half (45.2%) of experts expected a shift to a high carbon mobility mode [48]. Nonetheless, many people still had to use public transport because they did not have alternative modes [48]. Additionally, a resurgence of the COVID-19 pandemic is expected until 2024 and even further [48]. Moreover, during the COVID-19 pandemic, several people changed their perceptions, favoring the use of sustainable mobility modes to protect the climate [54]. Thus, it is required for policymakers to make use of public funds both for improving public transport infrastructure (i.e., Mobility-as-a-Service (MaaS)) and for supporting the technological advancement of green vehicles (i.e., shared autonomous electric and alternative-fuel vehicles) to make our societies highly sustainable (or safe and trusted) in the era of building back with the Paris Agreement and the Sustainable Development Goals (SDGs). These efforts should concentrate on five primary targets—road safety, energy efficiency, sustainable infrastructure, urban access, and reduced fossil fuel subsidies [47, 48, 52, 55]. The direct impact of the COVID-19 pandemic and demographic changes caused by an aging population must be regarded as a basis for the transition toward green and healthy sustainable transport and for increased investment in public transport to match new requirements [49, 54]. [Page 26, line 701-716 in revised manuscript] As discussed in subsection 1.1.5, the impact of the COVID-19 pandemic has encouraged people to get back into their cars for their commute rather than using public transportation (i.e., bus and train) because of safety concerns [49, 52, 54]. At the same time, more people changed their perceptions and preferred using sustainable mobility modes to protect the climate [54]. In order to meet the current and future needs, the strategy committee must consider an increase in national R&D funding in some technical fields, including Cluster 1 (Core electric/electronic sub-assembly for green cars) and Cluster 3 (Hydrogen charging infrastructure), to accelerate the electrification of public transport faster than the targeted year. Owing to increased energy consumption and congestion in the urban environment, increased government investment in public transport services and adoption of a connected and integrated system across mobility modes, including active and shared mobility solutions (Cluster 4. Autonomous vehicle service platform) are strongly recommended. In addition, it is required for the committee to regain the public’s confidence in public transport by more investment such as safer and cleaner trains and buses. The new R&D agenda for this new societal requirement needs to be discussed. |
Point 8: The discussion section is not correctly explored. In fact, the discussion and conclusions section is more related to conclusions than discussion. I recommend the division into two separate sections: discussion; conclusions. The authors need to better describe and explore the discussion section.
Response 8: The reviewer raises an important point. We appreciate this feedback from the reviewer. We followed the reviewer’s recommendations and have described the “Discussion and conclusion” sections in three separate sections – “4.1. Discussion for collaborative trans-regional R&D strategy on the future mobility”, “4.2. Conclusion”, “4.3. Limitations and Further Research” – of the revised manuscript according to the reviewer’s recommendation (please see below) [Page 25-27, line 651-780].
[Page 25-27, line 651-780] 4. Discussion and Conclusion 4.1. Discussion for collaborative trans-regional R&D strategy on future mobility The proposed framework for a collaborative trans-regional R&D strategy provides a variety of information to implement the two goals of reducing emission (Korean New Deal) and regional inequality (Region New Deal) in terms of regional, technological, and organizational dimensions. It is developed by considering the important functions of the systematic framework — an evidence-based situation analysis on particular sectors and technologies, a base for investment monitoring, and process management for inclusive communication among stakeholders [51]. In order to demonstrate the utilization of the framework, we established four research questions (RQ) (2 sub-research questions in RQ 2 and RQ3). First, based on RQ1, we revealed the overall status of regional government investment in future mobility from the perspective of automotive company locations during 2015–2020. This information may be a good starting point for a range of stakeholders to discuss the appropriateness of national R&D investment for the improvement of regions in the context of reliable sources and objective analysis. Second, based on RQ2-1 and RQ2-2, we presented the distribution and trends of investment in future mobility-related technology areas during 2015–2020. Third, based on RQ3-1 and RQ3-2, we showed various pieces of information to understand the comprehensive competitiveness of future mobility-related fields from the viewpoint of regions. For the central and local stakeholders, the investment information based on both the seven classified future mobility-related fields and the 17 regions in this study can be adopted as a medium that facilitates communication among them to discuss leading local R&D organizations for regional specialization. Finally, based on RQ4-1 and RQ4-2, we indicated the key investing local organizations and their R&D activities in future mobility-related technology areas, including academia, industry, and research institutes. This information may serve the potential trans-regional collaborative R&D partners or eligible experts and members of the strategy committee due to its objective, transparent nature, and legitimacy of the selection process. …… 4.2. Conclusion The Korean government recently proposed a new industrial strategy called the Korean New Deal, which combines climate change mitigation (Green New Deal) and the elimination of economic inequality (Regionally Balanced New Deal) to ensure sustainable development in response to the COVID-19 pandemic. Many scholars have asserted that a strategy for the green economic transition requires a long-term commitment to public support and funding for green R&D and innovation [27]; this agrees with the need for a fine-tuned analytical framework that considers individual regional variations caused by differences in the endowment of relevant assets such as natural resources, technologies, qualifications and skills, and institutional factors [8, 27, 33]. However, there are no practical examples of such a framework for industrial strategies that support the exchange of knowledge and other assets within and beyond regions by establishing horizontal and vertical coordination among the various stakeholders, actor networks, and policy agents at different spatial scales (from local to central government) [33]. To the best of our knowledge, this study represents the first empirical attempt to provide a framework for a precise trans-regional innovation scheme with regional, technological, and organizational dimensions. …… 4.3. Limitations and further research Despite these contributions, this study also has some limitations that present challenging questions for future research. One concern is that only nationally funded research project data from the central government were utilized. Although R&D expenditure determined by individual local governments is less than that of the central government, local government-led research project data also exist.…… |

Reviewer 4 Report
Dear authors,
your draft manuscript addresses a topic of current interest, is well structured, and provides sound results and conclusions. In my view, its main strength is that you provide solutions to real problems in the real world.
For these reasons, I'm going to recommend its publication.
Congratulations.
Author Response
Author's Response to the Reviewer 4’s comments:
Comments and Suggestions for Authors:
Dear authors, your draft manuscript addresses a topic of current interest, is well structured, and provides sound results and conclusions. In my view, its main strength is that you provide solutions to real problems in the real world. For these reasons, I'm going to recommend its publication. Congratulations.
Response 1: We sincerely appreciate the reviewer’s comments, their expert review, and the positive assessment of our manuscript. We are very honored by the opportunity to be published in this journal.

Round 2
Reviewer 3 Report
I appreciate the excellent review work done by the authors and the answers provided which are very well documented.